# Distinct functional constraints driving conservation of the cofilin N-terminal regulatory tail

Joel A. Sexton [1], Tony Potchernikov[2], Jeffrey P. Bibeau[2], Gabriela Casanova-Sepúlveda[2], Wenxiang Cao[2], Hua Jane Lou[1], Titus J. Boggon [1,2], Enrique M. De La Cruz [2] & Benjamin E. Turk [1] ✉

Cofilin family proteins have essential roles in remodeling the cytoskeleton through filamentous actin depolymerization and severing. The short, unstructured N-terminal region of cofilin is critical for actin binding and harbors the major site of inhibitory phosphorylation. Atypically for a disordered sequence, the N-terminal region is highly conserved, but specific aspects driving this conservation are unclear. Here, we screen a library of 16,000 human cofilin N-terminal sequence variants for their capacity to support growth in *S. cerevisiae* in the presence or absence of the upstream regulator LIM kinase. Results from the screen and biochemical analysis of individual variants reveal distinct sequence requirements for actin binding and regulation by LIM kinase. LIM kinase recognition only partly explains sequence constraints on phosphoregulation, which are instead driven to a large extent by the capacity for phosphorylation to inactivate cofilin. We find loose sequence requirements for actin binding and phosphoinhibition, but collectively they restrict the N-terminus to sequences found in natural cofilins. Our results illustrate how a phosphorylation site can balance potentially competing sequence requirements for function and regulation.

Phosphorylation is perhaps the most common reversible post-translational protein modification and has widespread roles in regulating cell behavior. Sites of phosphorylation evolve rapidly, in part because kinase recognition motifs can be introduced by a small number of mutations in flanking residues[1,2]. However, regulatory sites of phosphorylation are necessarily under additional sequence constraints beyond those required for kinase recognition. For example, intracellular environments harbor many protein kinases—up to ~500 for human cells[3]. Accordingly, just as the sequence surrounding the site must be tuned to its cognate kinase, there is also selective pressure to evade phosphorylation by other kinases[4–6]. Furthermore, regulatory phosphorylation often occurs in functionally important regions, so that variability in the local sequence may be limited to maintain basic function[7,8]. Finally, the specific sequence context of a phosphorylation

site may dictate how phosphorylation modifies protein behavior, for example, by generating a binding site for a phosphopeptide interaction module[9]. These factors are potentially in competition with one another, but there is currently limited insight into how they are balanced in any given phosphorylation site.

One example of a phosphoregulatory event occurring at a highly conserved site involves proteins in the cofilin/ADF (actin depolymerizing factor) family (hereafter referred to collectively as cofilin). Cofilin is a small actin-binding protein important for dynamic regulation of the actin cytoskeleton in eukaryotes and essential for multiple actin-dependent processes, including cytokinesis, cell polarity, migration, neurite extension, and mechanosensing[10–12]. Cofilin binds cooperatively to actin filaments, promoting severing at junctions between bare and decorated segments, as well as

[1]Department of Pharmacology, Yale School of Medicine, New Haven, CT 06520, USA. [2]Department of Molecular Biophysics and Biochemistry, Yale University, New Haven, CT 06520, USA. ✉e-mail: ben.turk@yale.edu

depolymerization, to produce new filament ends and a pool of actin monomers for reassembly[13–22].

Due to its major impact on the actin cytoskeleton, cofilin activity is tightly regulated by several mechanisms. Principal among these is phosphorylation at Ser3 within its disordered N-terminal region, catalyzed in most animals and some protists by members of the LIM kinase family (LIMK1/2 and TESK1/2)[23]. The N-terminal tail is an essential part of the actin-binding interface, and Ser3 phosphorylation inactivates cofilin by dramatically decreasing its affinity for actin[24]. The phosphomimetic cofilin$^{S3D}$ mutant binds to filamentous actin weakly but with higher cooperativity than the wild-type (WT) protein, and it severs filaments poorly, even when bound[24–26].

The N-terminus of animal cofilins contains a conserved acetyl-Ala2-Ser3-Gly4-Val5 sequence (by convention, we use residue numbering based on the unprocessed precursor)[11]. Orthologs from other kingdoms vary in the sequence upstream of Ser3 and may have other β-branched or aliphatic residues in place of Val5, but the phosphoacceptor residue and Gly4 are invariant. Factors driving the conservation of this sequence are currently unclear. Aside from full N-terminal deletion mutants that are non-functional, only the phosphomimetic S3D and phosphorylation-resistant S3A variants have been examined with respect to their effects on actin in vitro or in cells[24–28].

Recognition of cofilin by LIMK occurs through a docking interaction between the cofilin globular domain and a site adjacent to the kinase catalytic cleft on the kinase C-terminal lobe[29,30]. Consequently, the N-terminal region projects into the LIMK active site in a non-canonical orientation that lacks complementary binding interactions typically observed in other protein kinase-substrate complexes. Limited mutational analyses have suggested loose sequence requirements surrounding the phosphorylation site, save for an unusually strict exclusion of Thr as a phosphoacceptor residue[30]. Attempts to identify a LIMK phosphorylation site sequence motif using peptide libraries have been unsuccessful, presumably because the docking interaction is essential for efficient phosphorylation[29,31].

A platform for the evaluation of large numbers of N-terminal sequence variants would facilitate the discovery of those features important for the various aspects of cofilin functionality. Here we exploit the widespread requirement for cofilin to establish a budding yeast system for high-throughput evaluation of sequence variants. *S. cerevisiae* lacking their sole cofilin ortholog, Cof1, are inviable[32,33], but ectopic overexpression of mammalian cofilin-1 can rescue growth[24,29,34]. In this context, we have reported that inducible expression of human LIMK1 suppresses growth in a manner dependent on Ser3 phosphorylation[29]. With this system, we can evaluate in parallel the impact of a mutation on either core cofilin function or its regulation by LIMK based on its ability to support growth in the absence or presence of LIMK induction, respectively. Here, we have applied the system to screen a comprehensive collection of N-terminal sequence variants and subsequently conducted biochemical analysis of a panel of mutants. We found unexpectedly loose sequence requirements for each of the various aspects of cofilin function and regulation, with each dominated by its own distinct position within the N-terminal sequence. Overall conservation of the N-terminal region thus appears to be constrained by the collective requirements imposed by the full complement of cofilin functionality. These observations may have general implications for how regulatory phosphorylation sites emerge and are maintained through evolution.

## Results
### A dual-functionality cofilin library yeast competitive growth screen
To define cofilin N-terminal sequence requirements for actin severing and phosphoregulation, we leveraged the capacity of mammalian cofilin to substitute for yeast Cof1 in supporting growth[24]. We used a

yeast strain in which the endogenous *COF1* gene was placed under the control of a tetracycline-repressible promoter (TeTO$_7$-*COF1*)[35]. This strain grew poorly in the presence of doxycycline (DOX) due to loss of endogenous Cof1 expression, but ectopic expression of either yeast Cof1 or human cofilin-1 rescued growth (Fig. 1a). Our previous work with a temperature-sensitive *cof1* mutant strain demonstrated that ectopic expression of the LIMK1 catalytic domain (LIMK1$^{CAT}$) completely blocked growth in the context of human cofilin-1[29]. Likewise, we found that inducible expression of LIMK1$^{CAT}$ inhibited the growth of TeTO$_7$-*COF1* yeast expressing human cofilin-1 in a manner dependent on its phosphorylation at Ser3 (Supplementary Fig. 1). Expression of full-length (FL) LIMK1, which has lower cofilin kinase activity than LIMK1$^{CAT}$, caused only partial growth suppression (Fig. 1b). We chose to use FL LIMK1 for our screens as lower activity should allow us to better distinguish cofilin mutants that impact LIMK regulation to varying degrees. We next constructed a saturation mutagenesis library of the human cofilin-1 N-terminal region in which Ala2, Gly4, and Val5 were combinatorially randomized as all twenty potential canonical amino acids (Fig. 1c). Ser3 was maintained as either a serine or threonine residue to preserve a potential phosphorylation site, producing a library with 16,000 ($20^3 \times 2$) unique sequences. The library was introduced into a constitutive yeast expression plasmid with an N-terminal His$_6$ tag included to prevent potential sequence-dependent post-translational modifications (e.g., initiator methionine removal and acetylation) and to help normalize levels of expression. The His$_6$ tag did not prevent cofilin from rescuing yeast growth, severing actin, or being phosphorylated by LIMK1 (Fig. 1a, Supplementary Fig. 2). TeTO$_7$-*COF1* yeast harboring the inducible LIMK1 expression plasmid were transformed with the cofilin library and expanded in liquid media with a portion of the culture reserved as a starting point sample. The remaining cells were grown in raffinose to derepress the LIMK1 expression construct and treated with DOX to repress Cof1 expression. The culture was then split and propagated in either glucose or galactose to repress or induce LIMK1 expression, respectively (Fig. 1d). At various times, the cultures were sampled for plasmid DNA extraction. The variable library sequence within the recovered plasmids was PCR-amplified and subjected to next-generation sequencing. This analysis indicated how the relative abundance of each component of the library changed over time (Fig. 1e, Supplementary Data 1 and 2). In the context of competitive growth, DOX + glucose conditions should enrich for viable cells expressing functional library variants. Growth in DOX + galactose should deplete the culture of functional cofilin variants that could be phosphorylated and inhibited by LIMK1. We first analyzed our data from the DOX + glucose condition to identify core determinants of cofilin function.

### Functional constraints on the cofilin N-terminal sequence in supporting growth
Yeast grown in glucose were analyzed for enrichment of cofilin sequences that could presumably bind and sever actin filaments. We compared sequence representation between the beginning of culture in glucose and ~6 population doublings later (Figs. 1e and 2a), beyond which changes were less pronounced. We ranked each sequence by its average log$_2$ fold change in the normalized read count across three independently performed replicate screens (Fig. 2a). When analyzed this way, 4099 of the 16,000 library sequences were enriched in the population, suggesting that the majority of N-terminal substitutions were deleterious to cofilin function. As anticipated, native cofilin sequences performed well, including those from yeast (Arg-Ser-Gly-Val, rank 27), human (Ala-Ser-Gly-Val, rank 87), and *Arabidopsis* (Ala-Ser-Gly-Met, rank 262), though the most enriched sequence was non-native (Phe-Thr-Gly-Leu) (Fig. 2a, Supplementary Data 2). Examination of the 400 most enriched sequences overall identified a Leu-X-Gly-Met consensus motif. (Supplementary Fig. 3a). Within the N-terminal sequence, the

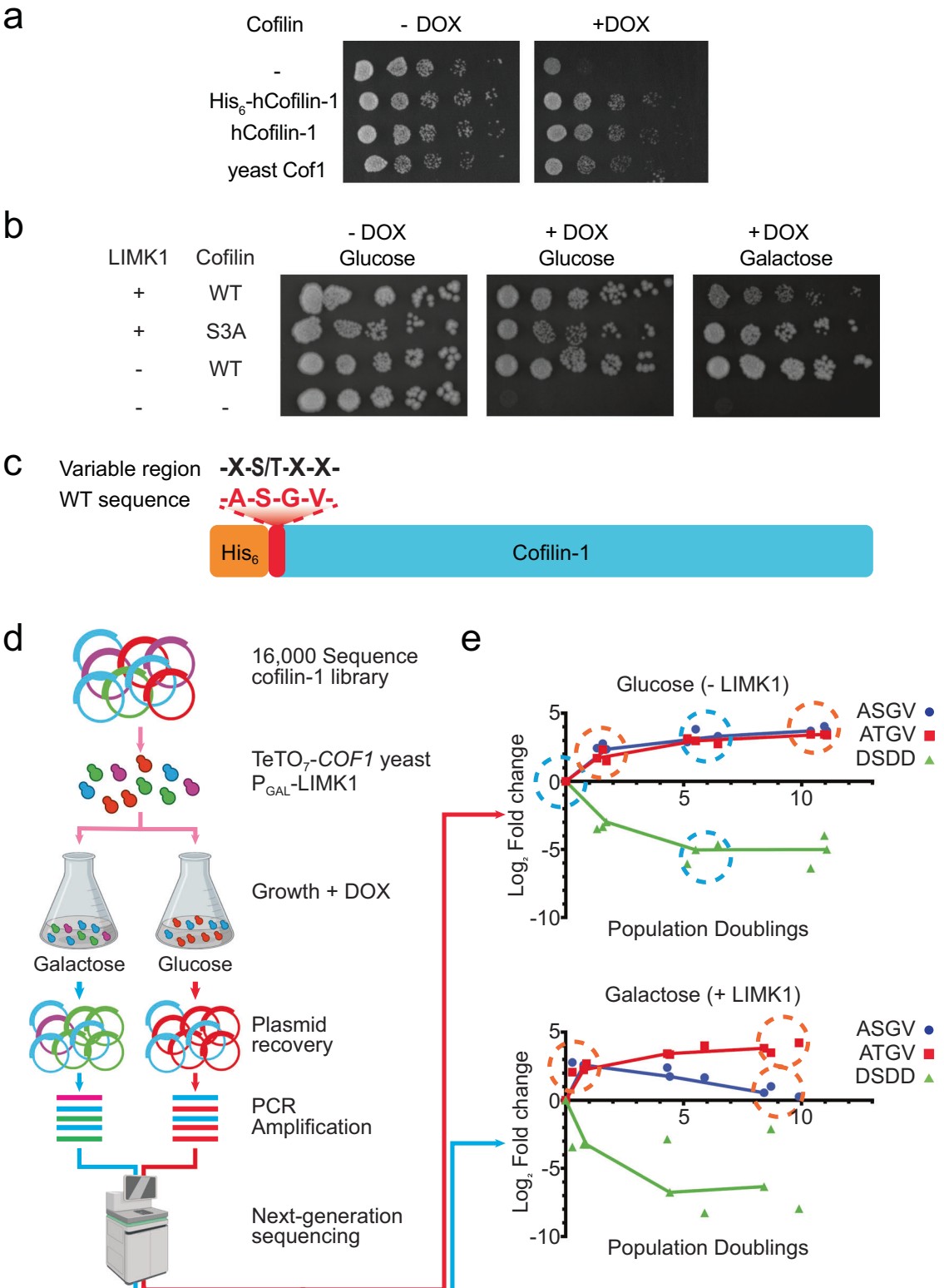

**Fig. 1 | A dual-functionality screen of a cofilin-1 N-terminal mutagenesis library.**
**a** Growth of TeTO$_7$-*COF1* yeast exogenously expressing the indicated cofilin in the presence or absence of doxycycline (DOX). Image representative of $n = 2$. **b** Yeast harboring the indicated expression plasmids were grown on either glucose or galactose to induce LIMK1 expression in the presence or absence of doxycycline. Image representative of $n = 7$. **c** Design of cofilin N-terminal combinatorial mutagenesis library. X indicates any of the 20 natural amino acids. **d** Schematic of the cofilin library dual-functionality screen. The schematic was made using Biorender.

**e** Graphs show a change in the relative representation of the indicated cofilin variants over time during culture in either glucose (no LIMK1) or galactose (LIMK1 induction). Data are from three replicate screens performed independently. Blue circles indicate timepoints used to identify functional sequences. Orange circles indicate timepoints used to identify sequences inhibited by LIMK1. Time zero is after the derepression of $P_{GAL}$-LIMK1 over the course of 1– 2 population doublings in raffinose + DOX media.

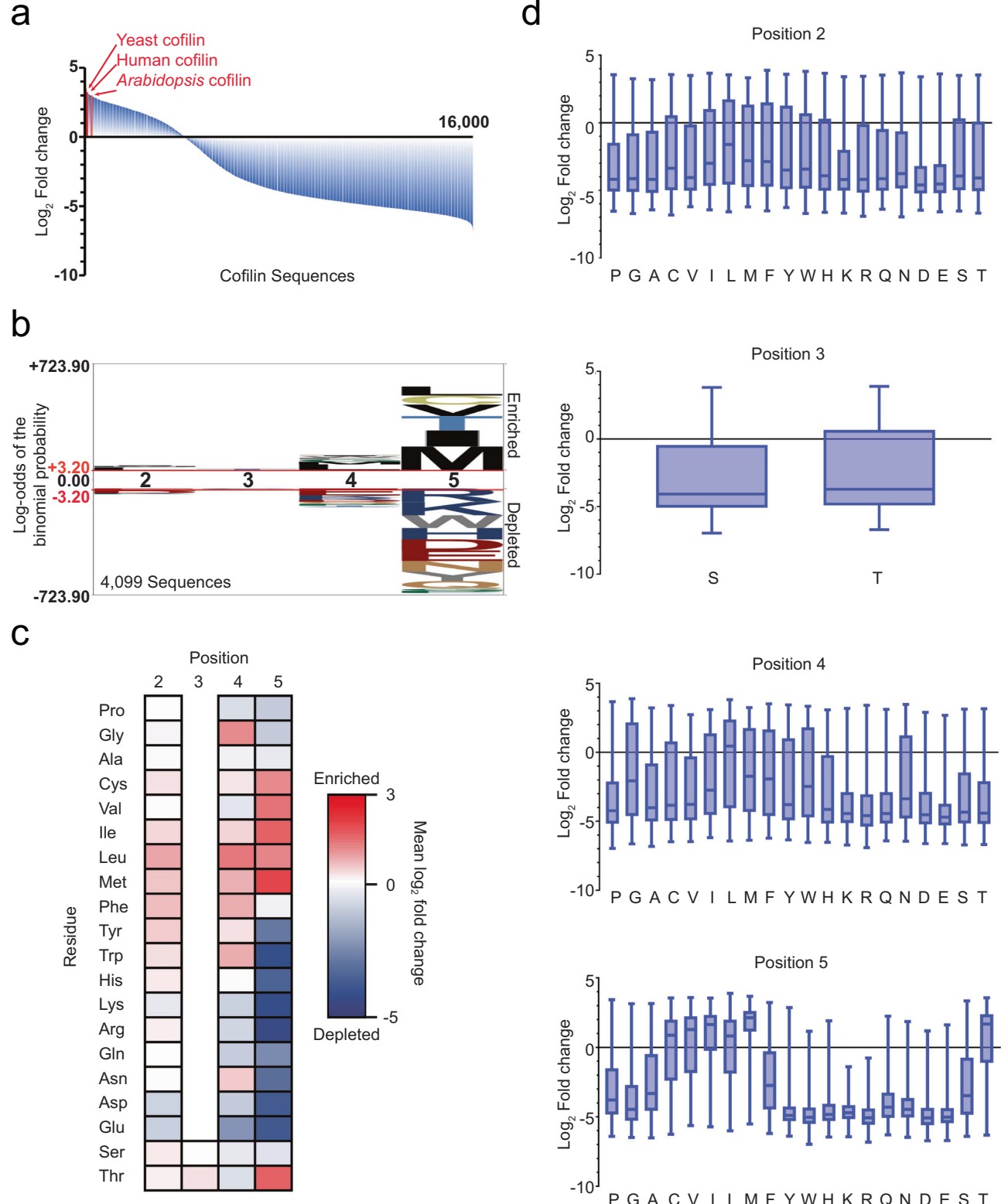

fifth position was the most stringently selective, with aliphatic or β-branched amino acids (73% of enriched sequences) being apparently required for function (Fig. 2b–d) and Met being most commonly enriched (17% of enriched sequences). By comparison, other positions within the sequence were less selective. Despite being highly conserved in plants and animals, the second residue (Ala2 in humans) was effectively indiscriminate, with modest enrichment for non-native hydrophobic over charged residues (Fig. 2d). Likewise,

enriched sequences had roughly equal representation of Ser (45%) and Thr (55%) at the phosphoacceptor (third) position. Finally, the selection at the fourth residue was intermediate. As found in all WT cofilin sequences, Gly was overrepresented at this position in enriched sequences (8% overall). Despite being absent from all native cofilin sequences, hydrophobic residues were generally selected at this position, with charged residues and most small residues being depleted.

**Fig. 2 | Cofilin library screen results for yeast growth rescue. a** Waterfall plot depicting the enrichment or depletion of all 16,000 cofilin library sequences expressed in TeTO$_7$-COF1 yeast grown in doxycycline. Sequences are ordered left to right from highest to lowest average enrichment from three independent screens. **b** Probability logo of the $n = 4099$ cofilin N-terminal library sequences enriched in the screen (average log$_2$ fold change from three independent experiments >0). The red line indicates the height threshold for the frequency of a residue in the enriched sequences being significantly different ($p = 0.05$) from the background frequency as calculated by a binomial probability function with Bonferroni correction. **c** Heat map of cofilin N-terminal library sequence enrichment or depletion. Color scale indicates the mean fold change of the 800 (positions 2, 4, and 5) or 8000 (position 3) library sequences containing the indicated residue across 3 independent replicates. **d** Distribution of cofilin sequences by enrichment scores according to the identity of residues at the indicated positions. Each distribution includes $n = 800$ sequences, with each value being the average from three separate experiments. Box plots indicate the median (middle line), 25th and 75th percentile (box), and 10th and 90th percentile (whiskers). Source data for all graphs is found in Supplementary Data 2.

Because our combinatorial library includes all possible combinations of residues at random positions, it has the potential to reveal cases where there is functional cooperativity between specific pairs of residues. In such cases, two residues would be co-enriched more frequently than expected if they behaved independently. We found many instances where residues appeared to co-vary in this manner. For example, Leu and Gly residues are represented similarly at position 4 within the 400 most enriched sequences. However, only Gly4 sequences were significantly enriched for a Val residue at position 5 (Supplementary Fig. 3a–c). Likewise, sequences with Val5 more prominently included Gly4 compared to those sequences containing Met5 (Supplementary Fig. 3d, e). The Gly4-Val5 pair, which is found invariably in animals and fungi, was uniquely over-represented within these most highly enriched sequences (Supplementary Fig. 3f) and may provide a fitness advantage over sequences combining independently favorable residues. The residue at position 4 likewise appeared to influence selectivity at position 2 in unexpected ways. For example, in the context of Gly4 but not Leu4, Glu2 was consistently preferred over a similar acidic Asp2 residue (Supplementary Fig. 3g). We furthermore noted distinct sequence preferences in the context of the two phosphoacceptor residues (Supplementary Fig. 4). Enriched Thr3 sequences in which the otherwise identical Ser3 sequence was depleted, for example, had a preponderance of charged residues at position 4 (Supplementary Fig. 4a, b). This phenomenon may reflect the capacity for Thr3 to better promote cofilin function, allowing it to compensate for the presence of these otherwise unfavorable residues. We also observed strong covariation of Thr3 with Thr5, as reflected in a large difference in mean log$_2$ fold change for Thr3/Thr5 variants compared with Ser3/Thr5 variants (1.7 vs. −0.28) (Supplementary Fig. 4c–e). While the basis for this and other examples of covariation are not immediately obvious, they are likely to promote either specific conformations or dynamics that enhance actin binding affinity or the capacity to sever actin once bound.

To confirm the results from the screen, we selected a number of mutants with varying degrees of enrichment for further analysis (Table 1). We initially assayed these mutants for their ability to support yeast growth on solid media, a format distinct from the competitive growth scheme used for the screen. Human cofilin-1 single or double-point mutants were expressed in TeTO$_7$-COF1 yeast and grown with or without DOX to control endogenous Cof1 expression. In keeping with their enrichment in the screen, cofilin$^{V5M}$ supported growth similarly to WT human cofilin (Fig. 3a), while yeast expressing cofilin$^{V5E}$ failed to grow. In contrast, Ala2 mutations had little effect on yeast growth, with both cofilin$^{A2L}$ and cofilin$^{A2D}$ supporting growth in the presence of DOX. Growth of yeast expressing the double mutant cofilin$^{A2L,V5M}$, which incorporates residues most highly enriched at each position, was indistinguishable from that of yeast expressing cofilin$^{WT}$ or cofilin$^{V5M}$. Substitutions to Gly4 displayed a range of growth phenotypes that generally correlated with behavior in the screen (Fig. 3b). Yeast expressing cofilin$^{G4L}$ grew similarly to yeast with cofilin$^{WT}$. Cofilin$^{G4A}$, which was slightly depleted in the screen, and cofilin$^{G4F}$, which was in the bottom half of enriched sequences, each supported growth albeit at slower rates than for cofilin$^{WT}$. The more substantially depleted variants, G4P, G4E, and G4K, provided, at best, a marginal amount of growth. Overall, we found that for most variants enriched in the

context of competitive growth in liquid culture, apparent differences in fitness did not translate into overt differences in growth rate on solid media. Furthermore, mutants that can support growth on solid media but at lower rates than cofilin$^{WT}$ were likely depleted in the screen due to being outcompeted by higher fitness variants. We note that the extent of growth rescue did not generally correlate with the level of cofilin expression (Supplementary Fig. 5). However, one of the most damaging mutations, cofilin$^{V5E}$, expressed at very low levels suggesting that it and perhaps other non-functional Val5 substitutions may destabilize the protein.

We next examined how the ability of a cofilin variant to support growth correlated with its biochemical behavior in vitro. We expressed and purified cofilin mutants using a system that produces untagged protein with the native N-terminal sequence (initiating with Ala2 or the desired substitution). We initially evaluated the panel of mutants for binding to actin filaments measured in a fluorescence quenching assay and further analyzed a subset of these mutants for their capacity to sever actin filaments in a TIRF microscopy-based assay. Phosphorylated cofilin$^{WT}$ and phospho-mimetic cofilin$^{S3D}$ were used as weak binding controls[25,28,36]. We used the cofilin concentration producing half-maximal occupancy ($K_{0.5}$) as a proxy for relative overall binding affinity[37]. Cofilin$^{WT}$ bound to actin filaments with a $K_{0.5}$ value of approximately 0.6 μM (Fig. 3c–e, Table 1), whereas cofilin$^{S3D}$ required ~8-fold higher concentrations to bind actin filaments. As anticipated, phosphorylated cofilin$^{WT}$ displayed almost no detectable binding at the concentrations tested. Almost all mutants that had supported growth on solid media, including the G4F, A2D, V5M, and the double mutant A2L/V5M, displayed actin binding comparable to cofilin$^{WT}$

### Table 1 | Properties of cofilin variants chosen for individual evaluation

| Cofilin variant | Growth on plate | Actin binding $K_{0.5}$, μM | $K_{0.5}$ 95% confidence interval μM | Screen rank | Log$_2$ fold change |
|---|---|---|---|---|---|
| WT | +++ | 0.61 | 0.54–0.67 | 87 | 3.19 |
| A2L | +++ | 0.53 | 0.48–0.57 | 182 | 3.05 |
| A2L,V5M | +++ | 0.85 | <0.93 | 197 | 3.02 |
| A2D | +++ | 1.5 | 1.4–1.6 | 822 | 2.55 |
| S3D | – | 4.7 | 4.4–4.9 | ND | ND |
| S3D,G4F | ++ | 1.06 | 1.02–1.09 | ND | ND |
| S3D,G4L | +++ | 0.39 | 0.37–0.41 | ND | ND |
| G4A | ++ | 3.2 | 3.0–3.3 | 4424 | −0.45 |
| G4L | +++ | 0.59 | 0.55–0.62 | 1260 | 2.33 |
| G4P | + | ND | ND | 5800 | −2.41 |
| G4F | ++ | 1.0 | 0.9–1.1 | 2292 | 1.76 |
| G4E | – | * | * | 9618 | −4.45 |
| G4K | – | 3.5 | 3.3–3.7 | 8153 | −3.98 |
| V5E | – | ** | ** | 10647 | −4.70 |
| V5M | +++ | 0.48 | 0.45–0.52 | 262 | 2.95 |

For growth on solid media: +++, equal growth in the presence or absence of DOX; ++, reduced growth on DOX; +, some detectable growth on DOX; –, no growth on DOX. For actin binding: *, none detectable; **, not assayed due to protein aggregation. ND not determined. Rank and log$_2$ fold change values for all variants are provided in Supplementary Data 2.

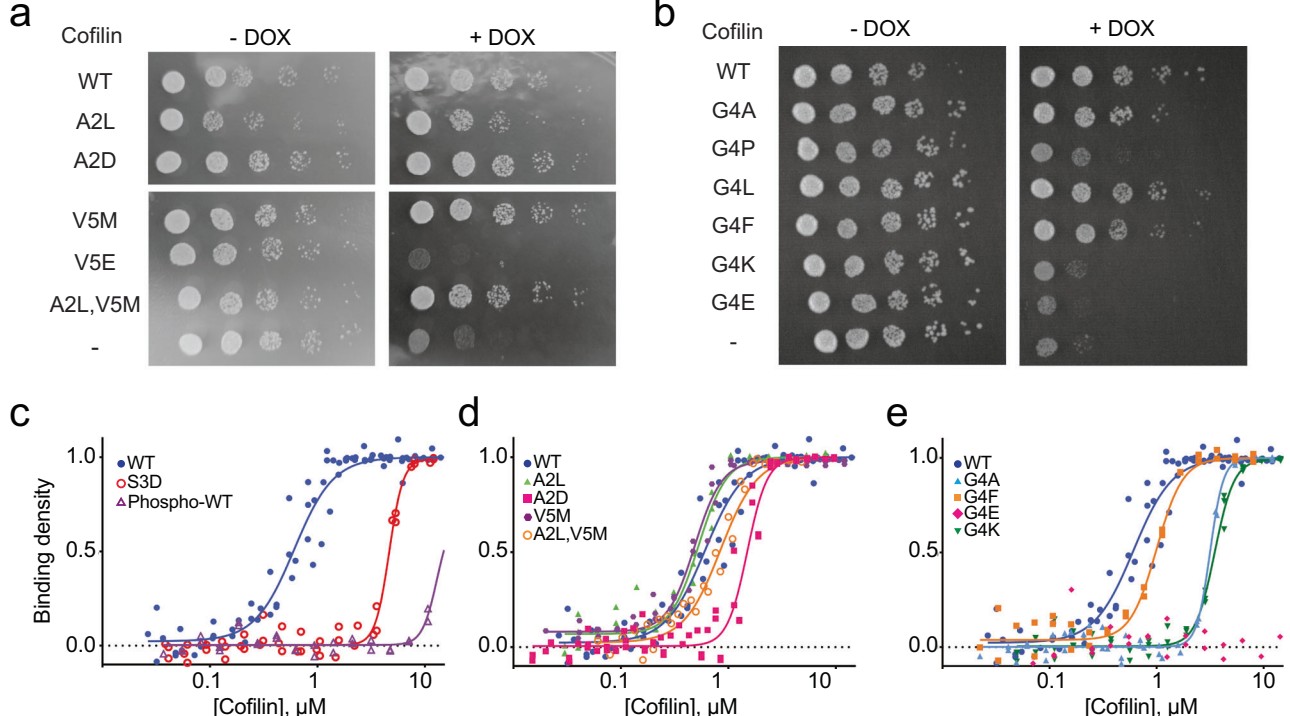

**Fig. 3 | Impact of cofilin N-terminal mutations on yeast growth and actin binding. a, b** Growth assay comparing cofilin mutants containing substitutions of residues Ala2, Gly4, or Val5. Images representative of $n = 3$ independent experiments. DOX, doxycycline. **c–e** Pyrene quenching assays comparing cofilin$^{WT}$ to the indicated cofilin mutants. Actin concentration pre-polymerization was $1\,\mu M$.

Independent biological replicates for binding assays are as follows: cofilin$^{WT}$ ($n = 4$), cofilin$^{A2L}$ ($n = 3$), cofilin$^{A2D}$ ($n = 4$), cofilin$^{S3D}$ ($n = 3$), phospho-cofilin$^{WT}$ ($n = 2$), cofilin$^{G4A}$ ($n = 3$), cofilin$^{G4F}$ ($n = 2$), cofilin$^{G4E}$ ($n = 2$), cofilin$^{G4K}$ ($n = 3$), cofilin$^{V5M}$ ($n = 3$), and cofilin$^{A2L,V5M}$ ($n = 2$). Source data are provided as a Source Data file.

(Fig. 3), and where tested, these mutants likewise severed actin filaments similarly to WT (Supplementary Figs. 6 and 7). Cofilin$^{G4A}$, which supported growth at a lower rate than cofilin$^{WT}$, showed somewhat weaker actin binding ($K_{0.5} = 3.2\,\mu M$), as did the G4K variant that failed to support growth ($K_{0.5} = 3.5\,\mu M$). Of all variants tested, only cofilin$^{G4E}$ showed essentially no binding to actin filaments (Fig. 3e), and accordingly, it had no detectable severing activity (Supplementary Fig. 6). Consistent with its low level of expression in yeast, cofilin$^{V5E}$ formed soluble aggregates when expressed in bacteria, and we were unable to isolate monomeric protein. Overall, we found a general correlation between yeast growth rate and actin binding, with a threshold $K_{0.5}$ of ~2 μM being required to support maximal growth.

### Identification of cofilin variants inhibited by LIMK1

To identify cofilin variants susceptible or resistant to LIMK1, we examined the relative abundance of functional variants in our screen following induction of the kinase by growth in galactose. We noted that depletion of the cofilin$^{WT}$ sequence from the culture lagged 1–2 population doublings behind the induction of LIMK1. We, therefore, followed variant representation from this time point onward over the course of 10 population doublings in the presence of either glucose or galactose (Fig. 1e), and we ranked sequences by the log$_2$ fold change in abundance. We found that approximately 1/3 of functional cofilin sequences (those enriched in glucose) became depleted in the presence of galactose, suggesting that they were phosphorylated and inhibited by LIMK1, with the remaining 2/3 persisting in the presence of galactose and thus presumably resistant to LIMK1 expression (Fig. 4a, Supplementary Fig. 8, and Supplementary Data 2). Functional sequences depleted in galactose overwhelmingly had Ser3 as the phosphoacceptor residue and were dominated by small residues at the Gly4 position, while conversely,

the majority of LIMK1-resistant sequences contained Thr3 and large hydrophobic residues (primarily Leu and Phe) in position 4 (Fig. 4b, c and Supplementary Fig. 8).

We initially examined the impact of phosphorylation site residue substitutions LIMK1-induced phosphorylation and growth inhibition in yeast (Fig. 5a). In contrast to yeast expressing cofilin$^{WT}$ (with Ser3), those expressing cofilin$^{S3T}$ were unaffected by LIMK1 induction and grew similarly to yeast harboring cofilin$^{S3A}$. Because our cofilin library did not include Tyr3 variants, we also examined the growth of yeast expressing cofilin$^{S3Y}$. While cofilin$^{S3Y}$ only partially rescued growth in the presence of DOX, growth was further inhibited by galactose-induced expression of LIMK1. Matching these observations, both cofilin$^{WT}$ and cofilin$^{S3Y}$ became phosphorylated upon LIMK1 induction (Fig. 5b) as judged by mobility shift on Phos-tag SDS-PAGE, while only a trace amount of cofilin$^{S3T}$ phosphorylation was observed. In similar experiments performed with yeast expressing the hyperactive LIMK1$^{CAT}$, cofilin$^{S3T}$ was still phosphorylated to only a small extent and fully supported growth (Supplementary Fig. 9). Consistent with these results and a recent report[30], LIMK1 phosphorylation of cofilin in vitro was blocked by either S3A or S3T mutation and was reduced but not eliminated when Ser3 was substituted with Tyr (Fig. 5c). LIMK2 was similarly unable to phosphorylate cofilin$^{S3T}$ (Supplementary Fig. 9d), indicating that this strong phosphoacceptor residue preference is conserved across isoforms. Collectively, these results confirm that while LIMKs are dual-specificity kinases in the sense that they can phosphorylate both Ser and Tyr, they are almost entirely devoid of activity on Thr residues.

Cofilin/ADF proteins are members of a larger family of actin-binding proteins having an ADF fold. Theoretically, other ADF domain proteins could be regulated by phosphorylation at sites analogous to Ser3. For example, the N-terminal region of twinfilin proteins, which consist of two ADF domains arranged in tandem, has

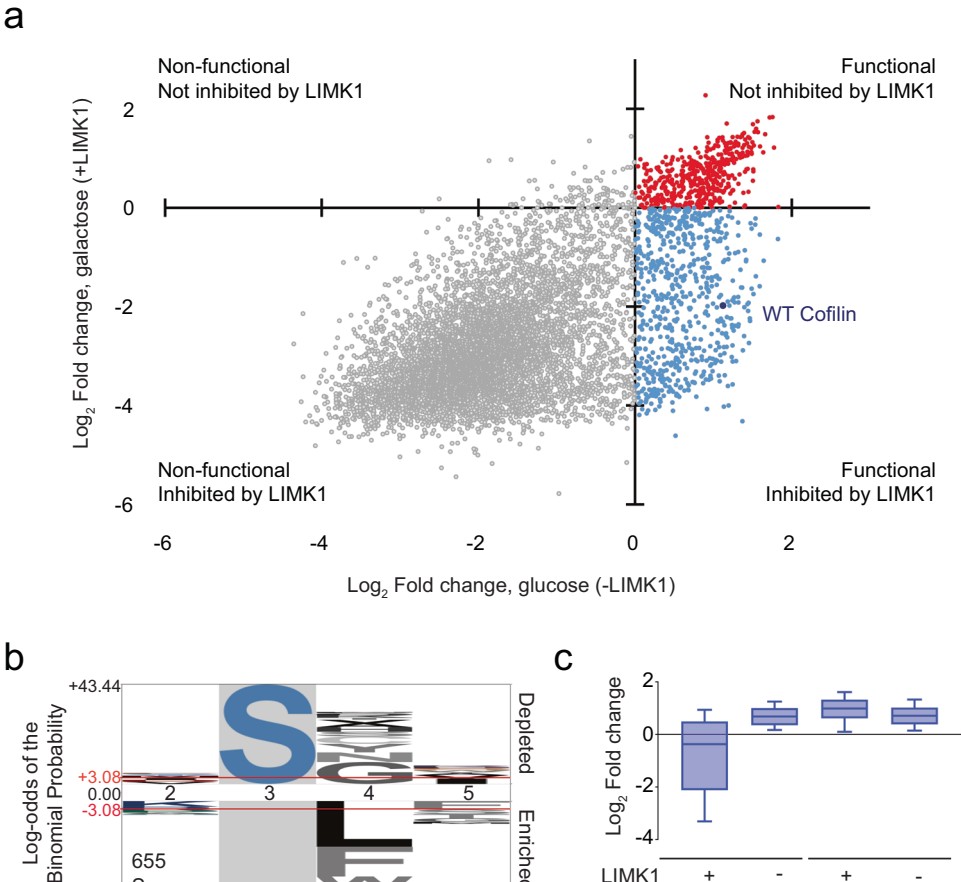

**Fig. 4 | Cofilin library screen results for yeast growth rescue with and without LIMK1. a** Scatter plot depicting the change in representation of Ser3 cofilin library sequences with and without LIMK1 expression. Data points show average enrichment in glucose and galactose across three independently performed replicates. **b** Logo of sequences supporting growth in yeast but are selectively inhibited by LIMK1 (lower right quadrant in panel **a**) on the background of all sequences that support yeast growth in the absence of LIMK1 (upper and lower right quadrants in panel **a**). **c** Distribution of cofilin sequences by enrichment scores with and without LIMK1 expression according to the identity of the phosphoacceptor residue. Each distribution includes $n = 8000$ sequences, and values are the average from three independent screens. Box plots indicate the median (middle line), 25th and 75th percentile (box), and 10th and 90th percentile (whiskers). Source data is found in Supplementary Data 2.

a potential phosphoacceptor residue at position 5, analogous to cofilin Ser3 (Fig. 5d). Notably, this residue is Thr in mammalian twinfilins, but is a Ser in the single ortholog from budding yeast, which lack LIMKs. Furthermore, key features of the LIMK docking region of cofilin centered on helix α5 are conserved to the N-terminal ADF domain of twinfilin (Supplementary Fig. 10)[29]. We therefore wondered whether the identity of the phosphoacceptor residue might serve to insulate twinfilin from phosphoregulation by LIMKs. We examined if LIMK1 could phosphorylate mouse or budding yeast twinfilin in vitro and whether phosphorylation was influenced by the identity of the putative phosphoacceptor. We found that LIMK1 robustly phosphorylated budding yeast twinfilin at ~40% of its cofilin phosphorylation rate yet had barely detectable activity on mouse twinfilin-1 (Fig. 5e). As anticipated, substituting Ser5 of yeast twinfilin with Thr abolished phosphorylation, confirming Ser5 to be the site of LIMK1 phosphorylation and suggesting flexibility in the identity of that residue in an organism lacking LIMKs. However, LIMK1 did not substantially phosphorylate mouse twinfilin-1^T5S, indicating a Ser phosphoacceptor residue is necessary but not sufficient for LIMK phosphorylation of ADF fold proteins. Collectively these results suggest that insulating twinfilin from inhibition by LIMK1 involves both the phosphoacceptor and additional structural features that prevent phosphorylation.

## Identification of phosphoregulatory constraints on the cofilin N-terminal sequence

In our screen, LIMK1-resistant variants were dominated by those having a Thr phosphoacceptor residue. However, Ser3 sequences having Leu, Phe, or Trp at position 4 also persisted despite LIMK1 expression, while Ser3 sequences depleted upon LIMK1 induction were enriched for Gly and other small residues at position 4 (Fig. 4b). We confirmed in growth assays on solid media that cofilin^G4L and cofilin^G4F, but not cofilin^G4A, protected yeast from LIMK1-induced growth inhibition (Fig. 6a, Supplementary Fig. 11). The most straightforward explanation for these results would be if variants with bulky hydrophobic residues at position 4 were poor substrates of LIMK1. However, we found instead that cofilin^G4L and cofilin^G4F became highly phosphorylated upon expression of LIMK1 in yeast, as did cofilin^G4A (Fig. 6b, c). Furthermore, LIMK1 phosphorylated cofilin^G4A, cofilin^G4L, and cofilin^G4F at rates faster than cofilin^WT in vitro, while cofilin^G4P was not detectably phosphorylated (Fig. 6d). A comparison of steady-state kinetic parameters confirmed that cofilin^G4F ($k_{cat} = 0.55 \pm 0.05$ s$^{-1}$, $K_M = 12 \pm 3$ μM) was a more efficient LIMK1 substrate than cofilin^WT ($k_{cat} = 0.20 \pm 0.03$ s$^{-1}$, $K_M = 9 \pm 3$ μM) over the full range of substrate concentrations (Fig. 6f, g). Collectively these observations suggest that the lack of cofilin^G4F and cofilin^G4L growth phenotypes in yeast expressing LIMK1 were not due to reduced phosphorylation.

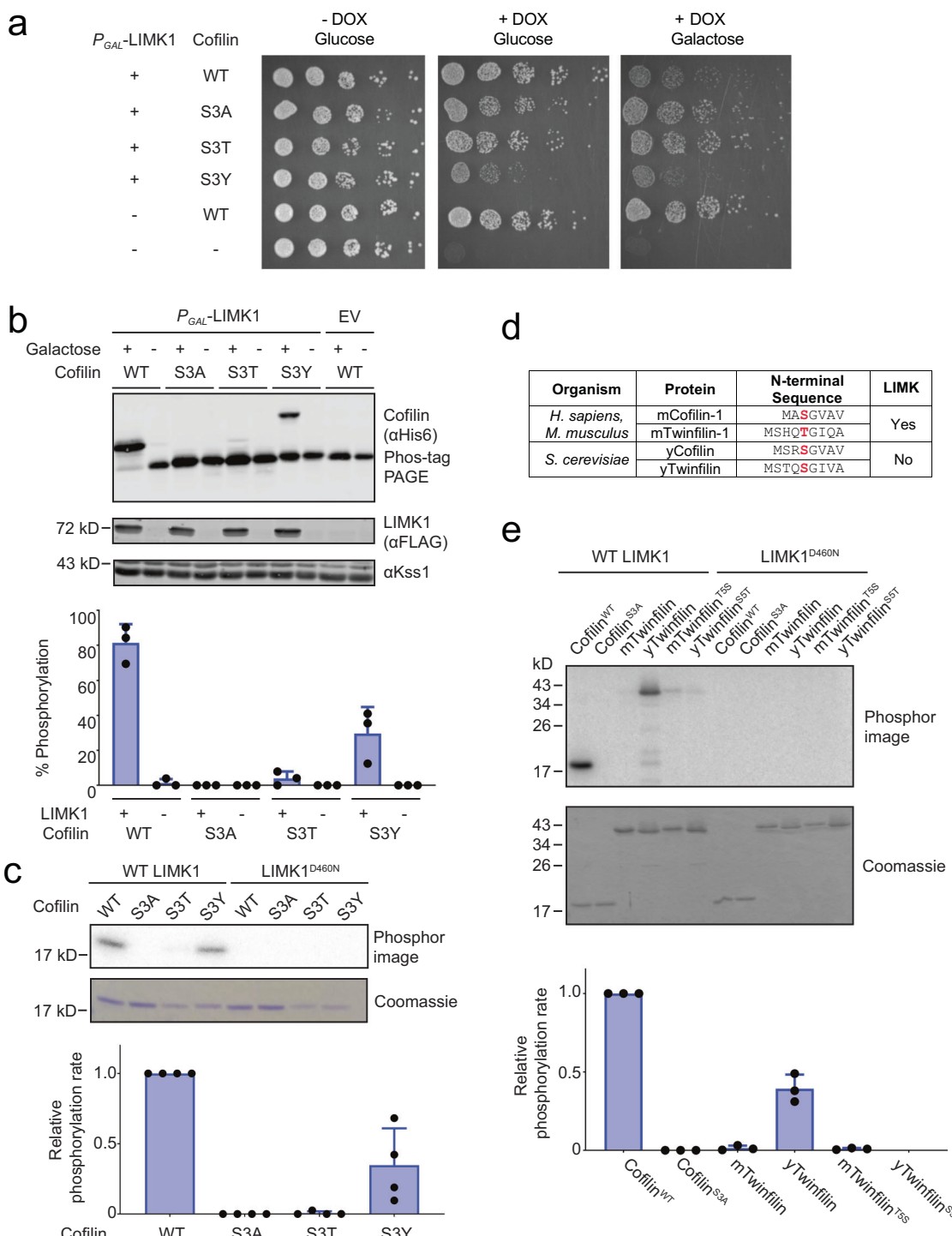

**Fig. 5 | Effects of cofilin Ser3 substitutions on phosphorylation by LIMK1.**
**a** TeTO7-*COF1* yeast expressing the indicated human cofilin-1 mutants were grown in the presence of doxycycline to repress endogenous Cof1 expression and/or galactose to induce LIMK1 expression. Images are representative of *n* = 3 independent experiments. DOX, doxycycline. **b** Immunoblots with the indicated antibodies of lysates corresponding to yeast plated in (**a**). Samples in the cofilin blot were separated on Phos-tag SDS-PAGE to resolve phosphorylated (upper band) from unphosphorylated protein (lower band). Kss1 serves as a loading control. The chart shows the fraction of cofilin phosphorylated, calculated from quantified intensities of the upper and lower bands. Data are presented as mean values ± SD for *n* = 3 independent experiments. **c** In vitro radiolabel kinase assay comparing

phosphorylation of purified cofilin (2 μM) by LIMK1^CAT (2 nM) for 10 min at 30 °C. The catalytically inactive LIMK1^D460N kinase domain serves as a negative control. The chart shows the quantification of phosphorylation rates of cofilin mutants normalized to cofilin^WT. Data are presented as mean values ± SD for *n* = 4 independent experiments. **d** N-terminal sequences of mammalian and yeast isoforms of cofilin and twinfilin. **e** In vitro radiolabel kinase assay showing phosphorylation of the indicated forms of cofilin-1 or twinfilin (2 μM) by LIMK1 kinase domain (2 nM) for 10 min at 30 °C. LIMK1^D460N kinase domain serves as a negative control. The graph shows the quantification of phosphorylation rates normalized to cofilin^WT. Data are presented as mean values ± SD for *n* = 3 independent experiments. Source data are provided as a Source Data file.

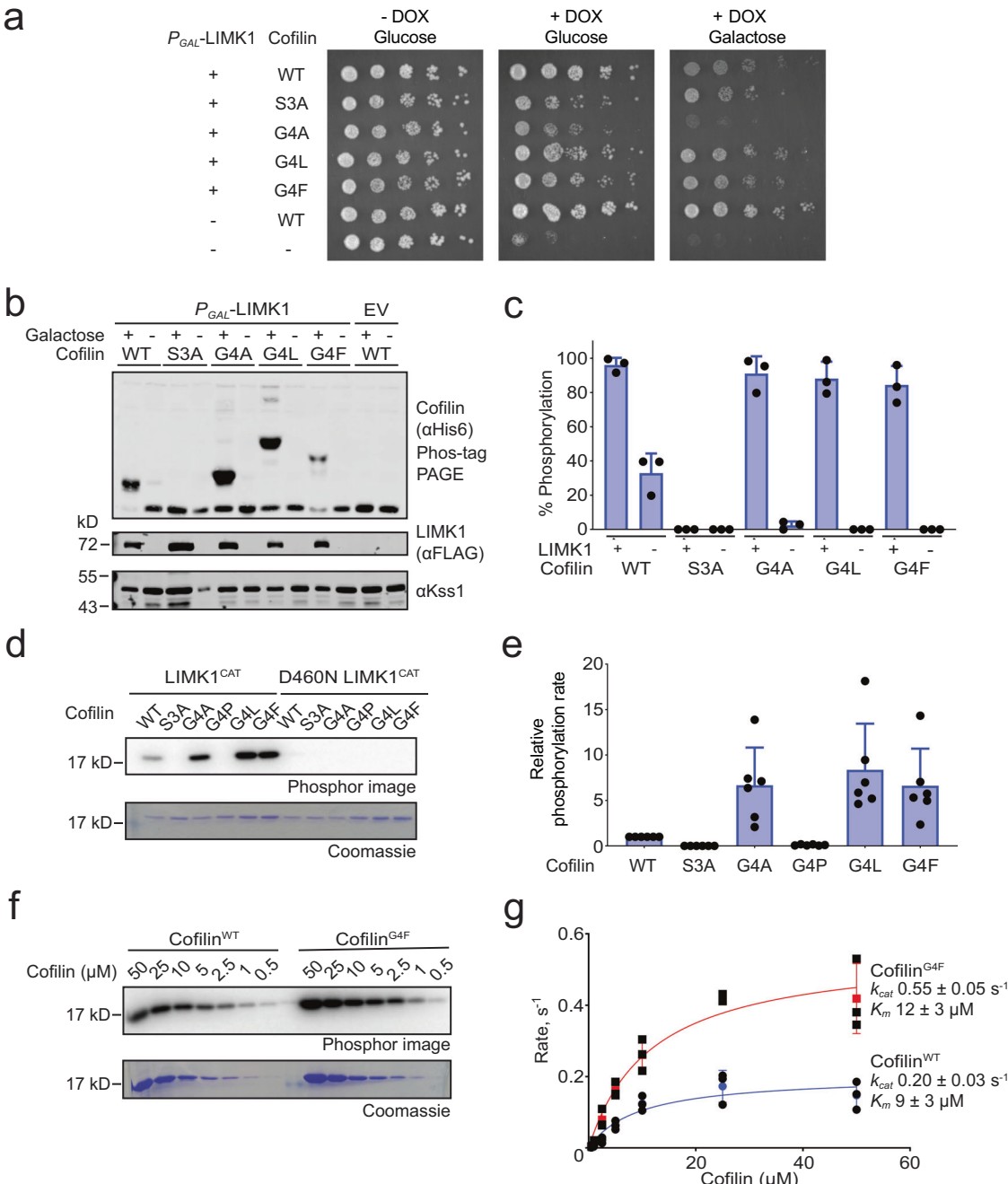

**Fig. 6 | Effects of cofilin Glycine 4 substitutions on LIMK1 substrate suitability.**
**a** Yeast growth assay with the TeTO$_7$-*COF1* strain expressing the indicated cofilin variants with or without LIMK1 expression. Representative of $n = 3$ independent experiments. DOX, doxycycline. **b** Immunoblots with the indicated antibodies of lysates corresponding to yeast plated in panel A separated by standard or Phos-tag PAGE. EV, empty vector. **c** Quantification of immunoblots measuring the fraction of total cofilin phosphorylated by LIMK1. Error bars show SD for $n = 3$ independent experiments. **d** In vitro radiolabel kinase assay comparing phosphorylation of purified 2 μM cofilin by 2 nM LIMK1 kinase domain for 10 min at 30 °C. The catalytically inactive LIMK1$^{D460N}$ kinase domain serves as a negative control. **e** Quantification of phosphorylation rates of cofilin mutants normalized to cofilin$^{WT}$. Error bars show SD for $n = 6$ independent experiments. **f** In vitro radiolabel kinase assay comparing phosphorylation of cofilin$^{WT}$ and cofilin$^{G4F}$ by 2 nM LIMK1 kinase domain across a range of concentrations. Reactions performed for 10 min at 30 °C. **g** Quantification of phosphorylation rates of cofilin mutants. Error bars show SD for $n = 3$ independent experiments. Kinetic parameters include SE for $k_{cat}$ and $K_m$. Source data are provided as a Source Data file.

As neither cofilin$^{G4F}$ nor cofilin$^{G4L}$ was impaired as a LIMK1 substrate, an alternative explanation for their ability to support growth in the presence of LIMK1 is that those mutations alleviate the inhibitory effect of Ser3 phosphorylation. To investigate this possibility, we used cofilin variants harboring a phosphomimetic S3D mutation to model constitutive phosphorylation. Consistent with its inability to sever actin, cofilin$^{S3D}$, as previously shown[24] was unable to support yeast

growth (Fig. 7a). In contrast, yeast expressing cofilin$^{S3D,G4F}$ or cofilin$^{S3D,G4L}$ grew at equivalent rates to those expressing their respective Gly4 single mutant. When examined in vitro, both cofilin$^{S3D,G4F}$ and cofilin$^{S3D,G4L}$ bound actin similarly to their non-phosphomimetic counterparts, effectively correcting the S3D substitution (Fig. 7b). Together, these results suggest that the presence of a large hydrophobic residue immediately downstream of the phosphoacceptor

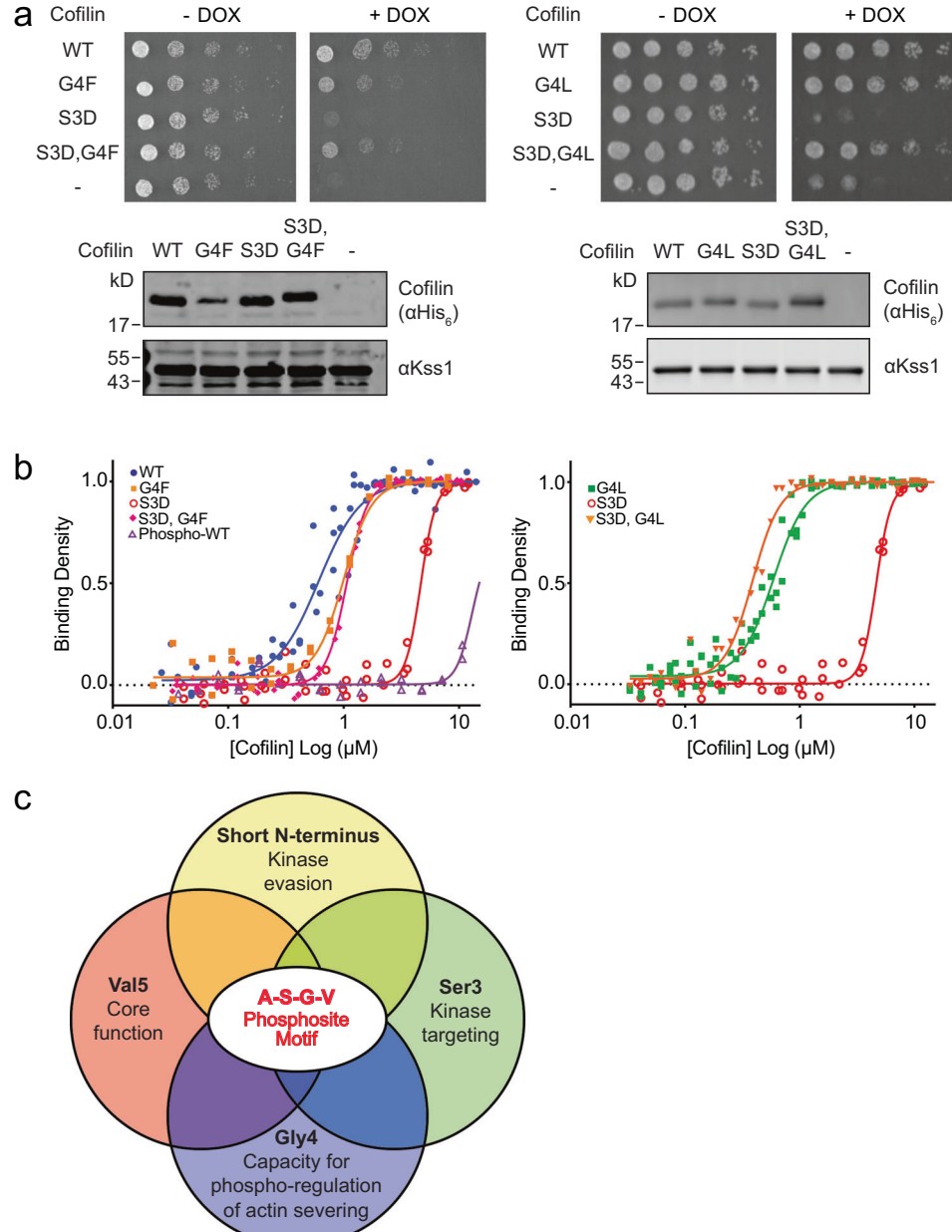

**Fig. 7 | Cofilin Gly4 mutations counter the inhibitory effects of phosphomimetic S3D substitution. a** Yeast growth assays with TeTO7-*COF1* strain expressing the indicated cofilin variants in the presence or absence of doxycycline (DOX). Immunoblots of cofilin protein levels are shown below, with Kss1 as a loading control. Images are representative of $n = 3$ independent experiments. **b** Pyrene quenching assays comparing cofilin^WT to cofilin variants with a phosphomimetic substitution or phosphorylation modification at residue Ser3. Actin concentration pre-polymerization was $1\,\mu M$. WT, S3D, and G4F single mutant data are the same as shown in Fig. 3 and are included here for comparison. Binding parameters are listed in Table 1. Independent biological replicates for binding assays are as follows: cofilin^WT ($n = 4$), cofilin^S3D ($n = 3$), phospho-cofilin^WT ($n = 2$), cofilin^G4F ($n = 2$), cofilin^S3D,G4F ($n = 3$), cofilin^G4L ($n = 6$), and cofilin^S3D,G4L ($n = 3$). **c** Overlapping constraints promoting conservation of the cofilin phosphosite motif sequence. Source data are provided as a Source Data file.

residue uncouples cofilin inhibition from phosphorylation. Accordingly, the absolute conservation of a Gly residue at position 4 may be due at least in part because it is required for phosphoregulation of actin-binding and severing.

## Discussion

Cofilin regulation of actin depolymerization is essential for normal cytoskeletal function[10–12]. Much of the functional regulation of cofilin converges on its N-terminal site of phosphorylation, yet how this short four amino acid stretch simultaneously impacts its relationship with its regulators, the LIM kinases, and its effector, actin, has not been explored. We used a yeast-based dual-functionality screen of a cofilin N-terminal sequence library and subsequent functional analysis to simultaneously determine the fitness of all possible combinations of N-terminal residues and to assess the importance of each of these amino acids. We identified features required for the maintenance of cofilin core functionality, targeting of cofilin phosphorylation by its kinases, evasion of non-cognate kinase phosphorylation, and phosphoregulation of its ability to bind and sever actin. We found that each aspect of cofilin functionality is mediated by distinct sequence features of its N-terminal sequence, which collectively restrain sequence variation across the entire tail (Fig. 7c).

This system allowed us to simultaneously determine the fitness of all possible combinations of N-terminal residues. Previous

                    

mutagenesis library screens have similarly used yeast growth as a readout for kinase activity and docking interactions[38–40]. Other approaches, such as phage display or droplet-based screening, could theoretically allow high-throughput analysis of actin binding, with a potential advantage of consistent levels of expression and a single readout[41]. However, it would be difficult to conduct large-scale in vitro screens examining actin severing or phosphorylation by LIMK. Some potential advantages of our approach are that all interactions occur in a eukaryotic cellular environment, that selection requires actin severing (and not simply binding) activity, and that we are able to simultaneously screen for cofilin function and regulation by LIMK.

Given the high degree of conservation of the cofilin N-terminal sequence, we were surprised to find relatively loose constraints on the sequence for maintaining sufficient activity to support growth. The deleterious effect of most substitutions was largely due to selectivity for aliphatic and β-branched amino acids at the fifth position. This range of residues is reflected in various cofilin orthologs, including those from mammals (Val), *Arabidopsis* (Met), maize (Leu), *Acanthamoeba* (Ile), and starfish (Thr)[24,42,43]. In all three-dimensional structures of cofilin solved to date, the sidechain of this residue contacts the cofilin globular domain, suggesting that it may stabilize the protein (Supplementary Fig. 12). Indeed, cofilin[V5E] aggregated readily during expression and purification. The suitability of variants at other positions was not readily predicted from known structures. For example, despite allowing interaction with actin, the presence of bulky residues in place of Gly4 is incompatible with the actin-binding mode observed by cryo-EM and would thus require an alternative conformation. By contrast, a more conservative G4A substitution was deleterious. Substitutions to Ala2 were most tolerated overall and had little effect on cofilin severing activity. This residue is absent from *Acanthamoeba* actophorin and is thus dispensable for actin binding[44]. We conjecture that the conservation of small residues at this position may be important for maintaining N-terminal processing that would be dispensable in the context of our screen. We note that in our viability-based screen, variants promoting the fastest growth would not necessarily be those that bind actin with the highest affinity and that a hyperactive cofilin could be deleterious for growth. However, when examined individually, mutations that imparted fitness defects consistently reduced actin binding, and there was a general correlation between rank in the screen and affinity. Finally, we note that many variants capable of supporting growth did appear to have fitness defects evident in a pooled competitive growth format, but a large number were indistinguishable from the WT sequence.

Our finding that almost all cofilin variants with a Thr3 phosphoacceptor residue were resistant to LIMK1 expression and our subsequent biochemical analysis agree with a recent report that cofilin[S3T] could not be phosphorylated by LIMK[30]. The few Thr3 cofilin sequences selectively depleted by LIMK1 were strongly enriched for Tyr residues at the adjacent position (~40% of sequences, Supplementary Fig. 13). As LIMK1 can efficiently phosphorylate cofilin[S3Y], it would seem likely that these variants are phosphorylated on Tyr4 rather than Thr3[30]. Ser/Thr kinase phosphoacceptor specificity is canonically determined by the identity of the residue immediately downstream of the DFG motif in the kinase activation loop (DFG + 1), which appears to promote a conformation facilitating phosphate transfer[45]. However, due to the non-canonical nature of the interaction, cofilin Ser3 is too distant from the LIMK DFG + 1 residue for it to impart substrate specificity[29]. Furthermore, the relative openness of the LIMK active site and the flexibility of the cofilin N-terminus suggest that steric clashes with the Thr3 sidechain are unlikely to preclude phosphorylation. Alternatively, LIMK and cofilin[S3T] may form a non-productive complex that sequesters the phosphoacceptor from the catalytic center. The majority of Ser/Thr kinases across species prefer Ser over Thr, potentially explaining why Ser3 is conserved even in organisms lacking LIMK[46].

In addition to optimizing a substrate for its cognate kinase, a phosphorylation site sequence may also be tuned to avoid the large number of other kinases present within the cell. For example, autophosphorylation sites in the human epidermal growth factor receptor appear to be suboptimal as a means to avoid phosphorylation by multiple non-receptor Tyr kinases[5]. As most kinases make extended interactions on either side of the phosphosite, the location of Ser3 so close to the cofilin N-terminus may facilitate evasion of non-cognate kinases[47]. By contrast, LIMK-cofilin binding does not require these canonical interactions, as demonstrated by its ability to phosphorylate the N-terminal Ser1 of actophorin[43]. Notably, the extended N-terminal sequences of yeast and plant cofilin orthologs may facilitate a more canonical mode of interaction in organisms lacking LIMKs. By a similar token, because it lacks selective pressure to evade LIMKs, yeast twinfilin can be an efficient LIMK substrate, which we could attribute to its Ser phosphoacceptor. Because LIMK-induced growth suppression appears entirely reversed in the context of a cofilin S3A mutation, twinfilin phosphorylation does not appear to confer a growth defect, but its potential phosphoregulation under other conditions warrants further investigation. Though conservation of Thr at the N-terminus of mammalian twinfilin does not appear necessary to evade phosphorylation by LIMK, it may contribute to its insulation from other pathways.

Save for the exclusion of Pro residues, LIMK1 was promiscuous with respect to the sequence surrounding the cofilin phosphorylation site, consistent with a prior report examining a small panel of mutants[48]. Indeed, LIMK-resistant sequences having bulky hydrophobic residues in place of Gly4 were phosphorylated faster by LIMK1. Our observation that G4F and G4L substitution can correct the phosphomimetic S3D mutation in vitro and in cells suggests that Gly4 is critical for cofilin to be regulated by phosphorylation. We postulate that this phenomenon underlies the widespread conservation of Gly4 in all cofilins. How might Gly4 optimize phosphoregulation? In molecular dynamics (MD) simulations, Ser3-phosphorylation or S3D mutation increases the dynamics of the N-terminal region and compromises its docking to actin[25]. A flexible Gly4 residue could be required for phosphorylation to increase dynamics, which may contribute substantially to the undocking of the N-terminus. Alternatively, if the hydrophobic binding site for the N-terminal tail repels a charged phospho-Ser3, the sidechain of Leu or Phe could recapitulate the N-terminal binding interaction if bound in a distinct conformation. It has also been suggested that phospho-Ser3 participates in intramolecular interactions with one or more basic patches within the cofilin globular domain to sequester the N-terminus away from actin[49,50]. In this scenario, reducing the flexibility of the N-terminal sequence by Gly4 substitution might preclude these intramolecular interactions, leaving the N-terminus free to interact with actin through the canonical binding mode. Further research will be required to distinguish between these models. Overall, our studies reveal specific determinants of cofilin regulation and how it interacts with and severs actin filaments.

## Methods

### Plasmids

The high-copy constitutive yeast expression vector (pRS423-GPD) for His[6]-tagged human cofilin-1, the bacterial expression vector for N-terminally His[6]-SUMO-tagged human cofilin-1 (in pEBDuet28a), and the galactose-inducible yeast expression vectors (pRS415-GAL1) for N-terminally FLAG-tagged LIMK1[CAT] and full length LIMK1 were previously described[29]. The cofilin-1 library and all cofilin mutants expressed in yeast using the pRS423-GPD vector were His[6]-tagged (MASHHHHHHGAGA) N-terminal to the initiator methionine. The bacterial expression vector for N-terminally His[6]-tagged human cofilin-1 was generated by excising the SUMO coding sequence from pEBDuet28a-His[6]-SUMO-cofilin-1 by Gibson assembly. The high-copy

yeast expression vector for yeast Cof1 was generated by shuttling the genomic DNA fragment from pRS316-*COF1* (produced by Mark Hochstrasser's laboratory)[34] into the EcoRI site of pRS423. Bacterial expression vectors for GST-tagged mouse twinfilin-1 and yeast twinfilin were a gift from the Bruce Goode laboratory[51,52]. Point mutations in all plasmids were made by either QuikChange site directed mutagenesis or by Gibson assembly following amplification with mutagenic primers. Oligonucleotides used in this study for cloning and mutagenesis are provided in Supplementary Data 3.

To generate the cofilin-1 N-terminal mutagenesis library, an oligonucleotide pool spanning the region was synthesized with phosphoramidite trimer mixtures encoding all 20 amino acids at the sites encoding Ala2, Gly4, and Val5, and an ACC/AGC mixture (encoding a mix of Ser and Thr) in place of Ser3 (core DNA sequences encoding the four residues are provided in Supplementary Data 1, and the full sequence is provided in Supplementary Data 3). Oligonucleotides were PCR amplified and inserted into pRS423-GPD-His$_6$-cofilin-1 plasmid using the 5' BamHI restriction site upstream of the start codon and a 3' SalI restriction site that was introduced by silent mutagenesis within the cofilin-1 coding sequence. Electrocompetent DH10β *E. coli* (Invitrogen ElectroMAX) were transformed with ligation products, providing at least 1000 transformants per library variant. Plasmid library DNA was recovered from the pool of transformed bacterial colonies, PCR amplified, and sequenced on an Illumina NovaSeq instrument to confirm the full representation of all variants.

### Cofilin library screening

The *S. cerevisiae* TeTO$_7$-*COF1* strain[35] (Horizon Discovery TH_5610) was transformed sequentially with pRS415-GAL1-FLAG-LIMK1 and the cofilin library plasmid pool and frozen in aliquots to conduct replicate screens. For each screen, transformed yeast was pooled and grown at 30 °C with shaking in liquid culture with selective media (SC-His-Leu) containing 2% glucose at a starting OD$_{600}$ of 0.1. After 4–5 population doublings, a portion was diluted to an OD$_{600}$ of 0.1 with SC-His-Leu media containing 2% raffinose and 10 mg/L DOX. After the culture reached an OD$_{600}$ of 1–1.5, a portion was reserved for plasmid recovery (T$_0$ sample), and the remaining culture was split and diluted into SC-His-Leu liquid media containing 10 mg/L DOX and either 2% glucose or 2% raffinose/1% galactose to OD$_{600}$ = 0.1. Cultures were subjected to three growth and dilution cycles in which they were grown to an OD$_{600}$ of 1–2 and diluted back to an OD$_{600}$ of 0.1, saving a portion for plasmid recovery at each step (T$_1$–T$_3$ samples). After the screen was complete, plasmids were recovered from cell pellets using a QIAGEN Spin Miniprep kit. The N-terminal variable region of the plasmid pool was PCR amplified to attach sequencing adaptors and add barcodes, and sequenced on an Illumina NovaSeq instrument.

The relative representation of each cofilin variant at each timepoint was calculated by dividing the number of reads that variant by the total read count. The enrichment or depletion of a cofilin-1 library sequence in the absence of LIMK1 induction was calculated as the ratio of its relative representation at T$_0$ and T$_2$ in glucose. Enrichment or depletion following LIMK1 induction was calculated from timepoints T$_1$ and T$_3$ in glucose or galactose. Sequences were ranked by the average log fold changes in representation across three replicate screens. Data were processed using Microsoft Excel 16.75. Values in the heat map in Fig. 2 are the mean fold change of all library sequences containing the indicated residue at the position shown. Probability logos were generated using pLogo[53] with the indicated sequences as the foreground and the full library as the background. Logos indicate the probability that the frequency of a given residue at each position is different from the background frequency of all sequences in the library.

### Immunoblotting

Yeast lysates were fractionated by either conventional SDS-PAGE or Mn$^{2+}$ Phos-tag SDS-PAGE before transfer to polyvinylidene difluoride (PVDF) membranes. Phos-tag acrylamide gels were prepared with 12% acrylamide, 100 μM MnCl$_2$, and 50 μM Phos-tag Acrylamide (Fujifilm AAL-107). Membranes were blocked with 5% powdered milk in Tris-buffered saline solution with 0.05% Tween-20 (TBST) for 2 h at room temperature and incubated with primary antibodies diluted 1:2000 in TBST containing 5% milk overnight at 4 °C. Primary antibodies included: anti-Penta-His (QIAGEN, #34650), anti-FLAG M2 (Sigma-Aldrich, #F3165), and anti-Kss1 (Santa Cruz, #sc-6775-R). Membranes were washed with TBST before incubation with fluorophore-labeled anti-mouse IgG (LI-COR Biosciences, #D10603-05) or anti-rabbit IgG (Invitrogen, #A21109) secondary antibodies at 1:20,000 dilution in TBST containing 5% bovine serum albumin (BSA) for 1 h at room temperature. Membranes were washed with TBST and imaged with an Odyssey CLx imager (LI-COR Biosciences). Fluorescence signals were quantified using Image Studio 5.2.5 software.

### Yeast growth assays

To evaluate the ability of cofilin variants to support growth, overnight cultures of TeTO$_7$-*COF1* yeast transformed with the indicated expression vectors were diluted to OD$_{600}$ of 0.1 in selective media and grown in a shaking incubator at 30 °C until 3–4 doublings had occurred. Portions of each culture were reserved for immunoblotting, and a series of 5-fold dilutions (OD$_{600}$ range from 0.5 to 0.0008) were spotted onto SC-His agar plates with or without 10 mg/L DOX. Plates were incubated at 30 °C for 3–4 days before imaging. Lysates for immunoblotting were prepared by pelleting cells at 4700×*g* for 2 min and vortexing the cell pellet with glass beads in TCA extraction buffer (10 mM Tris, pH 8.0, 10% TCA, 25 mM NH$_4$OAc, 1 mM EDTA). Proteins were pelleted by centrifugation and resuspended in 100 mM Tris (pH 11.0) with 3% SDS before boiling for 5 min. Protein concentrations were determined by BCA assay, and equal amounts were separated by SDS-PAGE and subjected to immunoblotting as described above.

The effect of LIMK1 expression on cofilin phosphorylation and yeast growth was evaluated using TeTO$_7$-*COF1* yeast co-transformed expression vectors for the indicated cofilin variants and pRS415-GAL1-FLAG-LIMK1, pRS415-GAL1-FLAG-LIMK1$^{CAT}$ or empty pRS415-GAL1 vector as a control. Yeast was grown overnight in SC-His-Leu containing 2% raffinose to derepress the GAL promoter, diluted to an OD$_{600}$ of 0.1, and allowed to grow through 3–4 population doublings at 30 °C. A portion of the cells were spotted and grown as described above on SC-His-Leu agar plates containing 2% glucose, 2% glucose with 10 mg/L DOX, or 2% raffinose/1% galactose with 10 mg/L DOX. The remaining culture was split, and LIMK1 expression was either induced or repressed by adding galactose to 1% or glucose to 2%. After 12 h, equal OD$_{600}$ units of each culture were pelleted, and lysates were prepared by TCA extraction as described above. Lysates were fractionated by both conventional SDS-PAGE and Mn$^{2+}$ Phos-tag SDS-PAGE before immunoblotting.

### Protein purification

Human cofilin-1 variants were expressed in *E. coli* Rosetta(DE3) cells overnight at 16 °C by induction with 0.5 mM IPTG. Cells were pelleted, snap-frozen on dry ice/EtOH, thawed on ice, and resuspended in bacterial lysis buffer (20 mM Tris, pH 7.5, 140 mM NaCl, 1 mM DTT, 0.4% Igepal CA-630, 1 mM MgCl$_2$, 200 μg/ml lysozyme, 10 μg/ml pepstatin A, Roche cOmplete protease inhibitor cocktail, and 30 U/ml DNAse 1). Cells were lysed by sonication and pelleted by centrifugation (13,870×*g* for 30 min at 4 °C). The supernatant was extracted and incubated with TALON affinity resin (Takara Bio) while rotating at 4 °C for 1–2 h. The resin was transferred to a gravity flow column, drained, and washed once with 0.5% Igepal CA-630 in PBS and once in 10 mM imidazole, 20 mM Tris, pH 8.0, 140 mM NaCl. His$_6$-SUMO-tagged cofilin variants were washed an additional time with cofilin storage buffer (10 mM HEPES, 100 mM NaCl, 1 mM DTT, 10% glycerol) and eluted from the resin by overnight digestion with SUMO protease at 4 °C. N-terminally

His$_6$-tagged cofilin variants were eluted with imidazole (20 mM Tris, pH 7.5, 250 mM imidazole, 100 mM NaCl, 1 mM DTT) and dialyzed into cofilin storage buffer overnight at 4 °C. Cofilin used for actin binding and severing assays was further purified by size exclusion chromatography on a Superdex200 10/300 GL column in 25 mM Tris, pH 8.0, 0.5 mM DTT, 0.25 mM EGTA. Cofilin eluted as a single peak across multiple fractions, which were pooled and concentrated using Amicon Ultra centrifugal filter concentrators (3 kDa cutoff). Protein concentration was determined by A$_{280}$ absorbance using a Thermo Scientific 2000 nanodrop instrument.

GST-tagged yeast and mouse twinfilin variants were expressed in *E. coli*, and cells were lysed as described above. The clarified cell lysate was incubated with glutathione Sepharose resin for 2 h at 4 °C. The resin was washed once with 0.5% Igepal CA-630 in PBS and once with cofilin storage buffer. Proteins were eluted by overnight digestion with GST-tagged HRV 3C protease at 4 °C. Twinfilin concentration was determined by comparison to BSA standards following SDS-PAGE and Coomassie-staining.

FLAG-tagged human LIMK1$^{CAT}$ for in vitro kinase assays was purified using a yeast expression system. TeTO$_7$-*COF1* yeast transformed with pRS423-GPD-His$_6$-cofilin-1$^{S3A}$ (to support the growth of yeast during LIMK induction) and pRS415-GAL1-FLAG-LIMK1$^{CAT}$ were grown in selective liquid media (SC-His-Leu) containing 2% raffinose at 30 °C with shaking to an OD$_{600}$ of 2. LIMK1 expression was induced by the addition of galactose and nutrient-rich yeast extract peptone (YP) broth, bringing the final concentrations of media to 1×YP media and 1% galactose. Cells were induced for 12 hr. at 30 °C before cells were harvested, pelleted, washed once with water, and snap frozen. The cell pellet was thawed on ice and resuspended in yeast lysis buffer (50 mM HEPES, pH 7.4, 150 mM NaCl, 1 mM EDTA, 0.5% Triton X-100, 10% glycerol, 0.5 mM DTT, 10 µg/ml leupeptin, 2 µg/ml pepstatin A, 2.5 mM NaPP$_i$, 10 µg/ml aprotinin, 1 mM β-glycerophosphate and 1 mM Na$_3$VO$_4$). Cells were lysed by vortexing with glass beads at 4 °C. Cell lysates were pelleted, and the recovered supernatant was incubated with M2 anti-FLAG affinity resin in batch for 1–2 h at 4 °C. The resin was washed twice with yeast lysis buffer and twice with FLAG wash buffer (50 mM HEPES, pH 7.4, 100 mM NaCl, 1 mM DTT, 0.01% Igepal CA-630, 10% glycerol, 1 mM β-glycerophosphate, and 100 µM Na$_3$VO$_4$) before the protein was eluted with FLAG wash buffer containing 0.5 mg/mL 3×FLAG peptide (Sigma). LIMK1 concentration was determined by running alongside BSA standards on SDS-PAGE with Coomassie staining.

To generate Ser3-phosphorylated cofilin, the FLAG-LIMK1 catalytic domain was expressed in yeast and purified as described above but not eluted from the resin. Purified cofilin$^{WT}$ was incubated with resin-bound LIMK1 catalytic domain overnight at 4 °C in kinase reaction buffer (50 mM Tris, pH 7.5, 100 mM NaCl, 5 mM MgCl$_2$, 5 mM MnCl$_2$, 1 mM DTT) with 1 mM ATP. The resin was then pelleted, and the supernatant dialyzed overnight at 4 °C into 50 mM Tris, pH 6.8 with 1 mM DTT. Phosphorylated and unphosphorylated cofilin species were separated by cation-exchange chromatography using a MonoS 5/50 GL column. Cofilin phosphorylation was assessed by Mn$^{2+}$ Phos-tag SDS-PAGE.

### Actin filament binding assays

The binding affinity of cofilin variants for actin filaments was compared using in vitro pyrene quenching assays as previously reported[25,37,54]. Serial dilutions of purified cofilin variants were incubated with polymerized pyrene-labeled actin (1 µM, ~83% labeled) in 1× KMI buffer (20 mM imidazole, pH 7.0, 50 mM KCl, 2 mM MgCl$_2$, 2 mM DTT, 200 µM ATP, 1 mM NaN$_3$) for 1 hr at 25 °C. Pyrene quenching was measured in a plate reader (excitation 366 nm, emission scan 409 nm). Data were normalized to the signal for unquenched pyrene-actin alone and saturated cofilactin filaments, and they were fitted to a model-independent sigmoidal dose-response curve using GraphPad Prism 9. The cofilin concentration at half-maximal binding was defined as $K_{0.5}$.

### Actin severing assays

The severing activity of cofilin variants was assessed using TIRF microscopy as previously described[25]. Alexa-647 labeled actin filaments (5% labeled) were polymerized for 1 h in 1× KMI buffer. Serial dilutions of purified cofilin variants were incubated with the polymerized actin for 1 h before placement on poly-L-lysine coated glass slips and imaged directly by TIRF microscopy. The recorded images were skeletonized using ImageJ FIJI and analyzed with Persistence Software version 4.2.3 to quantify the actin filament lengths[55]. Persistence Software is available to download for free at delacruzlab.yale.edu/persistence-software.

### Radiolabel kinase assays

Purified cofilin or twinfilin (2 µM) was incubated with LIMK1 catalytic domain (2 nM) and ATP (12.5 µM with 0.1 µCi/µL [γ-$^{32}$P]ATP) in kinase reaction buffer (defined above) at 30 °C. At 5 min and 10 min incubation, aliquots were removed and quenched with 4x SDS-PAGE loading buffer. Samples were boiled for 5 min and fractionated by 15% acrylamide SDS-PAGE. Gels were Coomassie stained and dried before exposure to a phosphor screen. Phosphor imaging and quantification of radiolabel incorporation were conducted using a BioRad Molecular Imager FX Pro Plus with Image Lab software. Images shown are from the 10 min timepoint. Phosphorylation rates were calculated from the entire time course and normalized to that of wild-type cofilin-1. Michaelis-Menten parameters for cofilin$^{WT}$ and cofilin$^{G4F}$ were determined from reactions run at seven substrate concentrations ranging from 50 to 0.5 µM. To calculate absolute rates, a [γ-$^{32}$P]ATP standard curve (2.5, 5, and 10 nCi) was imaged alongside the Coomassie-stained gel. Reaction kinetics were calculated using Microsoft Excel 16.75 and GraphPad Prism 9.

### Reporting summary

Further information on research design is available in the Nature Portfolio Reporting Summary linked to this article.

## Data availability

Next-generation sequencing data generated in this study have been deposited in the Gene Expression Omnibus (GEO) repository under accession code GSE242403. The processed data are available in the Supplemental Data 1 and 2 files. Yeast strains and plasmids described in this study are available upon request from the corresponding author. Source data are provided in this paper.

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

## Acknowledgements

We thank Cameron Schmitz for technical assistance in generating cofilin mutants. This research was supported by the National Institute of General Medical Sciences of the National Institutes of Health under awards R01 GM102262 (to B.E.T. and T.J.B.) and R35 GM136656 (to EMDLC). J.S. and G.C.S. were supported by a T32 training grant from the National Institute of General Medical Sciences, and G.C.S. was supported by American Heart Association grants 835293 and 23DIVSUP1058562. TP was supported by the Yale College Dean's Office and the Hahn Scholars Program.

## Author contributions

J.A.S. and B.E.T. designed the study, and J.A.S, T.P, J.P.B., G.C.S, W.C., and H.J.L. performed experiments. J.A.S., H.J.L., and G.C.S. generated new reagents used in this study. J.A.S., T.P., J.P.B., G.C.S, W.C., T.J.B., E.M.D., and B.E.T. designed experiments, analyzed data, curated data, and prepared figures. J.A.S. and B.E.T. wrote the initial draft, with additional editing provided by T.P., W.C., T.J.B., and E.M.D.

## Competing interests

The authors declare no competing interests.
