## [Peer Review File · Nature Communications]

Reviewers' Comments:

Reviewer #1:

Remarks to the Author:

Cofilin is an essential protein required for rapid disassembly of the actin cytoskeleton across eukaryotic organisms. Phosphorylation of cofilin is the major mode of intracellular regulation of cofilin's function i.e. cofilin's ability to bind, sever and depolymerize actin filaments and in turn facilitate recycling of actin monomers. Lim kinase (LIMK) is the central player responsible for controlling cofilin's activity. LIMK phosphorylates cofilin at the Serine residue in the N-terminal disordered region (Ser3). LIMK-mediated phosphorylation drastically reduces cofilin's ability to bind and disassemble actin networks. While LIMK's role in tuning cofilin's activity is relatively well-appreciated, the role of N-terminal sequence of cofilin itself in specifying LIMK-cofilin relationship has remained poorly understood.

Here the authors screened a library of 16,000 human cofilin N-terminal sequences to identify sequence requirements for cofilin's ability to bind actin and be regulated by LIMK. The authors employ a combination of yeast growth assays as well as in vitro biochemical approaches to investigate how different cofilin variants influence yeast cell viability and severing of actin filaments.

I find the experiments to be well-designed, and the conclusions of the study are well-supported by the data presented data. The manuscript is timely, well-written and the study will be valuable to the field. I therefore recommend publication of the manuscript without any substantive changes.

Reviewer #2:

Remarks to the Author:

This work analyzes the roles of the disordered N-terminus in the actin-binding protein, cofilin. The N-terminal region of cofilin forms part of its actin-binding interface and is also the site of regulatory (inhibitory) phosphorylation by the kinase LIMK1. The current study uses a high-throughput approach, involving a competitive growth assay in yeast, to assess how variations in this short N-terminal sequence (spanning 4 aa) affect both the positive function of cofilin that is essential for growth and its susceptibility to inhibition by LIMK1. The implications are further explored via additional assays of actin binding and severing, plus tests of phosphorylation by LIMK1 in vivo and in vitro. The findings clearly reveal differing degrees of selectivity at each of the 3 positions surrounding the phosphorylation site, with residue p5 being most stringent. They also show a general agreement between actin binding strength and functional fitness during competitive growth. An especially intriguing finding (and one that would be very hard to predict) arose from the growth selection for variants that are resistant to LIMK1 inhibition; the most strongly resistant variants (with L, F, or W at p4) were not refractory to phosphorylation but instead were found to be refractory to the effects of phosphorylation. Another surprise is that although residue Gly4 is universally conserved in eukaryotes, this position tolerates multiple substitutions rather well; the possible explanations and implications finding seem worthy of explicit discussion. One minor weak point concerns the subject of "kinase evasion" (alluded to in the text and Fig 7), as it is unclear which findings pertain to this topic (if any).

Overall, this manuscript provides thought-provoking findings and valuable new insights for the field. The work is thorough and multifaceted, with high quality of data and analysis. I don't have any serious concerns or reservations about the work, although there are a variety of issues that I think would be valuable for the authors to elaborate upon or clarify in the text, as outlined below.

1. The Discussion, lines 329-330, claims "We identified features required for ... evasion of non-cognate kinase phosphorylation...", and the Venn diagram in Fig 7D implicates Ala2 in "Kinase evasion". This is perplexing. I could not identify any data that connect Ala2 to kinase evasion or any evidence that kinase evasion is dictated by the N-terminus. To the contrary, Fig 5E seems to suggest that the N-terminal sequence itself does not determine kinase susceptibility. These issues should be substantially clarified.

2. There seem to be additional interesting behaviors in the competitive growth dataset (in Dataset S1) that might be worth comment. Two examples:

(a.) The preferred residues at p2 seems to vary depending on the context of the remaining sequence. In particular, among p2 residues in the xSGM context, F and W are ranked #2 and #3, but F/W are the two worst residues in the context of xSGI. (Similarly, F is worst in xSGV.) Separately, in the xSGM context, the best and worst residues at p2 residue are E and D, respectively, which is curious given their chemical similarities. Perhaps the authors could comment on whether these features seem real versus within the range of noise/error, and if they can be sensibly interpreted.

(b) If the data are divided by the two possible residues at p3 (Ser vs. Thr), do the sequence combinations at the remaining 3 positions (p2, p4, p5) affect growth similarly in each setting? A simple scatterplot of the log₂fold values for Ser3 vs. Thr3 sequences suggests good agreement for combinations that yield either strong or weak growth, but remarkable disparities in the intermediate region (e.g., log₂fold values strongly negative for Ser3 but positive for Thr3); some examples include SsDT vs StDT, RsTC vs. RtTC, and LsRV vs. LtRV. These disparities might relate to the Thr3-Thr5 covariance illustrated in Fig S4A, but it seems a more extensive phenomenon. Some comment on how to interpret this behavior (and how to interpret the Thr3-Thr5 covariance already shown) seems worthwhile.

3. Regarding lines 173-175: "Gly4 sequences showed enhanced selection of Ile or Val residues at position 5, while Leu4 sequences were more likely to be co-enriched with Met5 (Fig. S3A-C)." Residues I/V/M are the top 3 in either context; are their different rank orders in Gly4 vs. Leu4 statistically significant (or reproducible in all 3 repeats)?

4. The Methods section is thorough but it lacks information about processing competitive growth data. It would be especially useful to describe the calculations used to generate the sequence logos (Figs 2B, 4B, etc.). In particular, do they reflect only the fractional representation of each residue within the set of sequences analyzed, or do they incorporate the actual magnitude of growth enrichment? It seems that the latter is used for the heatmap in Fig 2C, whereas I suspect the former is used for the logos, but I could find any confirmation of this. Related, the relevance of values on the logo y-axes (e.g., +3.20, +723.90, etc.) should also be explicitly stated.

5. Line 272: It is not clear what is being claimed or emphasized in Fig S10 B/C. The panel says "conserved residues (orange) important for LIMK1 binding ARE ALSO CONSERVED in Twinfilin-1..." Does this refer to conservation of residues BETWEEN Cofilin and Twinfilin? (If so, it is odd that K127 vs. E111, D122 vs A106, are included as "conserved".) Or, does it refer to conservation WITHIN/AMONG members of each homolog family (i.e, conserved among Cofilins, and also conserved among Twinfilins)? Also, is the conservation especially higher for this helix than for the remainder of the globular domain? Perhaps this issue would be better illustrated by a linear diagram of conservation across the entire domain rather than just this helix.

6. Line 196: I don't think these spot assays (in Fig 3A-B) can be translated into specific growth rates (e.g., "5-fold"). The spot densities likely reflects a mix of growth rate (affecting colony size) and survival (affecting colony number). It's probably better to just say growth was "slower", rather than claim a specific rate.

Minor points:

7. Line 191: "...the "consensus" double mutant cofilin A2L,V5M...". The meaning of "consensus" here is obscure; please explain in what sense this is a consensus.

8. Lines 237 and 240: Fractions don't need superscripts; e.g., 1/3 means one-third, there's no need to add "rd".

9. Line 245: I think this should cite Fig 4B,C (not 4A,B).

10. Lines 257-259: I think this should cite Fig 5C (not 5D). Also, clarify that this result refers to IN VITRO phosphorylation; otherwise, it sounds redundant with the in vivo findings in the preceding

sentence describing Fig 5B.

11. Line 269: In current Figure the relevant panel here is marked as 5D, not 5F. Moreover, in the Fig 5 legend the descriptions of panels D/E appear to be labeled as panels F/G.

12. In Fig 7C, the symbol for S3D G4F double mutant is different in the curve (diamond) versus the key (inverted triangle).

Reviewer #3:

Remarks to the Author:

In this manuscript, the authors have investigated N-terminal mechanisms of cofilin function and phospho-regulation by its cognate kinase, LIMK. They employ budding yeast and an elegant system to screen a randomized library of N-terminal cofilin mutants to identify sequence variants supporting actin regulation/depolymerization (indicated by growth in culture) and, concurrently, phospho-regulation by LIMK1.

The authors make some interesting findings and conclusions with overall outcomes that are broadly consistent with the current understanding of cofilin biology. Enriched variants align with native sequences. Variants that support growth in culture reflect capacities for actin binding and filament severing (with the later intermittently evaluated in this study). Val5 position confers the greatest selectivity for cofilin function and while Gly4 is invariant, this has intermediate selectivity for function. While residue 2 is invariably Ala, this seems to be most dispensable for function. LIMK phosphorylates serine3, can phosphorylate Tyr in this position, but not Thr which has been previously demonstrated as referenced by authors.

Most interestingly, Phe substitution of Gly4 does not prevent Ser3 phosphorylation but removes or negates its regulatory effects on cofilin activity. This is interesting as Ser3 phosphorylation status is widely associated with cofilin1 inactivation/regulation and this observation provides one biochemical context where these events are effectively uncoupled.

The experiments are well executed with convincing datasets that support the various conclusions. However, my main reservation for publication in journal such as Nature Communications is that main findings, as outline above, are either unsurprising or their broad significance in protein phosphoregulation unclear.

Hydrophobic residues at the position 4 removes phospho-regulation by LIMK is an interesting observation. But what is the broader significance given Gly4 is invariant in nature? The mechanistic basis that uncouples Ser3 phosphorylation from activity regulation is underdetermined but speculated to involve dynamic flexibility of N-terminal region. Whether this represents a generalized mechanism for other ADF/cofilins, LIMK substrates or in protein phosphoregulation more broadly is unclear. I'm also not entirely convinced that this explains the absolute conservation of Gly at 4. The residue at the 4 position has intermediate selectivity for supporting growth and actin regulation so some variability at this position could be expected. The speculated mechanisms in the discussion do not exclude another small residue (Fig 4B) that could similarly be conducive for cofilin function, LIMK1 phosphoregulation and flexibility of N-terminus. Conversely, Asp and Tyr at position 4 are charged or bulky residues that could similarly influence flexibility of the N-terminal region or the charged phosphate on Ser 3 but shown to support growth and are depleted with induction of LIMK (Fig. 4).

For completeness, the experiment in Figure 7 should incorporate G4L cofilin variant that is similar phosphorylated and rescues yeast growth in presence of LIMK. As phosphomimics do not fully recapitulate phosphorylation chemically or functionally (eg. Fig 3C), this reviewer is also left wondering whether actin binding/severing is similarly maintained by G4L and G4F mutants that are in vitro phosphorylated by LIMK1 (compared to a phosphomimic). One would expect it to be and easily confirmed given the established proteins and methods?

My other suggestion relates to the twinfilin expts which adds incrementally to phosphoacceptor site preferences for LIMK in Fig. 5 but seems also to present an opportunity to investigate

phosphoserine regulation of actin binding/severing properties of this conserved ADF protein (doi.org/10.1242/jcs.02860) and whether the flanking conserved Gly residue at +1 position has similar influence over the phosphoacceptor site as demonstrate in cofilin1.

Lastly, the manuscript is well written overall with exception Figure 5C-E are incorrectly labelled in the main text. Page 12 description of LIMK1 phosphorylation Ser3 substituted cofilin references Fig. 5D incorrectly. Similarly, references to twinfillin sequence and in vitro phosphorylation are mislabelled as Fig. 5F and 5G respectively. Figure 5 legends on page 35 also incorrectly indicate figure panels 5D and 5E as F and G.

Response to reviewers

J. Sexton, *et al.* "Distinct functional constraints driving conservation of the cofilin N-terminal regulatory tail"

We thank the reviewers for their positive comments on our manuscript and their constructive criticism. In response we have made changes to the text and figures, and we have conducted additional experiments. We believe these changes provide additional support for our conclusions and have greatly strengthened the manuscript.

Reviewer #1

Cofilin is an essential protein required for rapid disassembly of the actin cytoskeleton across eukaryotic organisms. Phosphorylation of cofilin is the major mode of intracellular regulation of cofilin's function i.e. cofilin's to ability to bind, sever and depolymerize actin filaments and in turn facilitate recycling of actin monomers. Lim kinase (LIMK) is the central player responsible for controlling cofilin's activity. LIMK phosphorylates cofilin at the Serine residue in the N-terminal disordered region (Ser3). LIMK-mediated phosphorylation drastically reduces cofilin's ability to bind and disassemble actin networks. While LIMK's role in tuning cofilin's activity is relatively well-appreciated, the role of N-terminal sequence of cofilin itself in specifying LIMK-cofilin relationship has remained poorly understood.

Here the authors screened a library of 16,000 human cofilin N-terminal sequences to identify sequence requirements for cofilin's ability to bind actin and be regulated by LIMK. The authors employ a combination of yeast growth assays as well as in vitro biochemical approaches to investigate how different cofilin variants influence yeast cell viability and severing of actin filaments.

I find the experiments to be well-designed, and the conclusions of the study are well-supported by the data presented data. The manuscript is timely, well-written and the study will be valuable to the field. I therefore recommend publication of the manuscript without any substantive changes.

We thank the reviewer for their positive comments on our manuscript.

Reviewer #2

This work analyzes the roles of the disordered N-terminus in the actin-binding protein, cofilin. The N-terminal region of cofilin forms part of its actin-binding interface and is also the site of regulatory (inhibitory) phosphorylation by the kinase LIMK1. The current study uses a high-throughput approach, involving a competitive growth assay in yeast, to assess how variations in this short N-terminal sequence (spanning 4 aa) affect both the positive function of cofilin that is essential for growth and its susceptibility to

inhibition by LIMK1. The implications are further explored via additional assays of actin binding and severing, plus tests of phosphorylation by LIMK1 in vivo and in vitro. The findings clearly reveal differing degrees of selectivity at each of the 3 positions surrounding the phosphorylation site, with residue p5 being most stringent. They also show a general agreement between actin binding strength and functional fitness during competitive growth. An especially intriguing finding (and one that would be very hard to predict) arose from the growth selection for variants that are resistant to LIMK1 inhibition; the most strongly resistant variants (with L, F, or W at p4) were not refractory to phosphorylation but instead were found to be refractory to the effects of phosphorylation. Another surprise is that although residue Gly4 is universally conserved in eukaryotes, this position tolerates multiple substitutions rather well; the possible explanations and implications finding seem worthy of explicit discussion. One minor weak point concerns the subject of "kinase evasion" (alluded to in the text and Fig 7), as it is unclear which findings pertain to this topic (if any).

Overall, this manuscript provides thought-provoking findings and valuable new insights for the field. The work is thorough and multifaceted, with high quality of data and analysis. I don't have any serious concerns or reservations about the work, although there are a variety of issues that I think would be valuable for the authors to elaborate upon or clarify in the text, as outlined below.

We thank the reviewer for their positive feedback on our manuscript. Our point-by-point response follows below.

1. The Discussion, lines 329-330, claims "We identified features required for ... evasion of non-cognate kinase phosphorylation...", and the Venn diagram in Fig 7D implicates Ala2 in "Kinase evasion". This is perplexing. I could not identify any data that connect Ala2 to kinase evasion or any evidence that kinase evasion is dictated by the N-terminus. To the contrary, Fig 5E seems to suggest that the N-terminal sequence itself does not determine kinase susceptibility. These issues should be substantially clarified.

We apologize for the confusion regarding the model shown in Figure 7D. The reviewer is correct that we have no evidence that Ala2 per se is involved in evading non-cognate kinases, nor does it contribute to LIMK phosphorylation site specificity. What we had intended to convey was because the N-terminal sequence has only one residue (Ala2) upstream of the phosphorylation site, Ser3 would almost certainly be disfavored by many other protein kinases, a point we make in the discussion section. Though there is no data in the manuscript that speaks to this specific point, there is ample evidence from prior work on kinase specificity that recognition of residues from the -5 to -2 positions is important for efficient phosphorylation by a number of kinases. To address this point, we have modified the kinase evasion sector of Figure 7D to indicate "short N-terminus" rather than "Ala2".

2. There seem to be additional interesting behaviors in the competitive growth dataset (in Dataset S1) that might be worth comment. Two examples:

(a.) The preferred residues at p2 seems to vary depending on the context of the remaining sequence. In particular, among p2 residues in the xSGM context, F and W are ranked #2 and #3, but F/W are the two worst residues in the context of xSGI. (Similarly, F is worst in xSGV.) Separately, in the xSGM context, the best and worst residues at p2 residue are E and D, respectively, which is curious given their chemical similarities. Perhaps the authors could comment on whether these features seem real versus within the range of noise/error, and if they can be sensibly interpreted.

We thank the reviewer for pointing out additional instances of context-dependent selectivity in our dataset. We agree that there are a large number of examples of this behavior. The divergent selection of Glu and Asp at position 2 is particularly interesting because it appears to occur largely in the context of a Gly residue at position 3. If we look broadly at the entire set of enriched sequences, we find that both Asp and Glu are significantly ($p < 0.05$ by pLogo analysis) deselected at position 2 (see the heat map in Fig 2C and variant distribution in Fig 2D). Consistent with the reviewer's comment, Glu2 is consistently more enriched than its corresponding Asp2 variant in the context of Gly4, shown below for sequences with favored residues at position 5 (Val, Ile, Leu, Met and Thr). These differences are statistically significant in pairwise comparisons ($p < 0.05$ by t-test) in all cases but one (xSGV). Notably, in the context of Leu4, none of the Asp2 sequences are significantly more enriched than their corresponding Glu2 sequences. Note that all of the variants shown here are enriched during the screen and thus can support growth, though at different rates. We agree that the differences in behavior between Asp and Glu sequences is difficult to rationalize and can only speculate that additional spacing afforded by the Glu sidechain limits repulsive interactions with the hydrophobic pocket of cofilin. In this regard, Leu4 may "correct" an Asp2 mutation as it does for the S3D phosphomimetic mutation, potentially by providing additional hydrophobic interactions to increase affinity with F-actin or by reducing conformational dynamics of the tail region. We have added additional content to the results section drawing attention to this phenomenon, and we include the panel below as Figure S3G.

Figure S3G. The graph shows the fold change in abundance of the indicated sequences following doxycycline treatment in glucose (source data is the same as for Fig 2A in the main text). Individual data points are shown from the three replicate screens. * indicates $p < 0.05$ by Welch's two-tailed t-test. All other differences between Glu and Asp variants in the same sequence context are not significant.

With respect to the context dependence of Phe2/Trp2 sequences, we believe the reviewer is in error, as our own analysis indicates that these residues are highly ranked for both xSGM and xSGI.

(b) If the data are divided by the two possible residues at p3 (Ser vs. Thr), do the sequence combinations at the remaining 3 positions (p2, p4, p5) affect growth similarly in each setting? A simple scatterplot of the log₂fold values for Ser3 vs. Thr3 sequences suggests good agreement for combinations that yield either strong or weak growth, but remarkable disparities in the intermediate region (e.g., log₂fold values strongly negative for Ser3 but positive for Thr3); some examples include SsDT vs StDT, RsTC vs. RtTC, and LsRV vs. LtRV. These disparities might relate to the Thr3-Thr5 covariance illustrated in Fig S4A, but it seems a more extensive phenomenon. Some comment on how to interpret this behavior (and how to interpret the Thr3-Thr5 covariance already shown) seems worthwhile.

We thank the reviewer for examining our data in a way we had not considered and drawing our attention to these phenomena. To examine broadly such position 3 context-dependent sequences, we considered pairs of sequences for that had discordant behavior between their corresponding Ser and Thr variants. These correspond to the upper left and lower right quadrants of the Ser vs Thr scatterplot described by the reviewer and reproduced below. As suggested by the reviewer, selected Thr sequences recapitulate the xTxT motif discussed in the original manuscript. At position 4, we find basic and acidic residues that are generally deselected in the context of the full library. By contrast, the selectively Ser-enriched sequences have Val4 and Phe5/Tyr5 residues overrepresented. These patterns are consistent with the reviewer's observations and indicate that indeed these context-dependent selections are common and statistically significant by pLogo analysis. One explanation that could account for some co-variation would be if there is a threshold level of cofilin activity required for maximal yeast growth. For example, it is reasonable to assume that in the context of the WT sequence, cofilin activity is not rate limiting for growth and thus an improved variant would offer no growth advantage. It is evident from Fig 2B and C that Thr3 is overrepresented in enriched sequences compared to Ser3. Thus if Thr3 generally promotes cofilin activity, its presence could simply compensate for suboptimal residues at other positions to reach the level of activity required for maximal growth. We do not have a straightforward explanation for specific instances of co-variation such as Ser3-Val4. We assume that these combinations somehow promote either specific conformations or dynamics that enhance affinity, which need not occur by increasing the cofilin-actin interaction surface.

We have added additional content to the results section describing this analysis, and we include the panels below in Figure S4 of the revised manuscript.

Figure S4A,B. Functional coflin sequences can depend on phosphoacceptor residue context. (A) Scatter plot showing enrichment of otherwise identical coflin sequences with either a Ser3 or Thr3 residue. Average log₂ fold change across the three replicate screens is shown. (B) Sequence logo of uniquely enriched Ser3 (top) or Thr3 (bottom) coflin variants.

3. Regarding lines 173-175: "Gly4 sequences showed enhanced selection of Ile or Val residues at position 5, while Leu4 sequences were more likely to be co-enriched with Met5 (Fig. S3A-C)." Residues I/V/M are the top 3 in either context; are their different rank orders in Gly4 vs. Leu4 statistically significant (or reproducible in all 3 repeats)?

The reviewer is correct that Val/Ile/Met are all the three most enriched residues at position 5 regardless of whether Leu or Gly is fixed at position 4. The important point being that only in the case of Gly4 is Val selected significantly ($p < 0.05$ that it is different from the background frequency, corresponding to log-odds > 3.08). For example, there are 23 occurrences of Gly4/Val5 in the top 400 sequences (log-odds = 9.64, $p = 1.4 \times 10^{-8}$) but only 11 occurrences of Leu4/Val5 (log-odds 2.28, $p > 0.05$). However, it is correct that Met5 is found with similar frequency in the top 400 Gly4 and Leu4 sequences. We have modified the results section when describing these observations, in particular removing the phrase suggesting co-enrichment of Leu4 with Met5.

4. The Methods section is thorough but it lacks information about processing competitive growth data. It would be especially useful to describe the calculations used to generate the sequence logos (Figs 2B, 4B, etc.). In particular, do they reflect only the fractional representation of each residue within the set of sequences analyzed, or do they incorporate the actual magnitude of growth enrichment? It seems that the latter is used for the heatmap in Fig 2C, whereas I suspect the former is used for the logos, but I could find any confirmation of this. Related, the relevance of values on the logo y-axes (e.g., +3.20, +723.90, etc.) should also be explicitly stated.

Sequence logos only represent enriched sequences on the (flat) background of all sequences in the library, and do not consider the magnitude of enrichment. We now describe explicitly in the figure legends which sequences were used to build the logo. The log-odds calculation is described in the paper reporting the pLogo method (O'Shea, et al. *Nat. Methods* 10, 1211-1212, 2013) and indicates the probability that the frequency of a given residue in the set of sample sequences is different from its frequency in the background set. The red line in the logo graph (i.e. 3.2 in Fig 2B) corresponds to a significant selection ($p = 0.05$). We now provide this definition in the figure legend and refer to the publication describing the calculations in the methods section.

The heat map in Fig 2C shows the mean enrichment score for all library sequences with a given residue at the indicated position and thus is related to the magnitude of enrichment. This quantitative definition is described in the figure legend and the heat map is accompanied by a scale bar. We have added content to the methods section describing how the heat map was generated.

5. Line 272: It is not clear what is being claimed or emphasized in Fig S10 B/C. The panel says “conserved residues (orange) important for LIMK1 binding ARE ALSO CONSERVED in Twinfilin-1...” Does this refer to conservation of residues BETWEEN Cofilin and Twinfilin? (If so, it is odd that K127 vs. E111, D122 vs A106, are included as “conserved”.) Or, does it refer to conservation WITHIN/AMONG members of each homolog family (i.e., conserved among Cofilins, and also conserved among Twinfilins)? Also, is the conservation especially higher for this helix than for the remainder of the globular domain? Perhaps this issue would be better illustrated by a linear diagram of conservation across the entire domain rather than just this helix.

The reviewer is correct that by “conserved” we meant that the interface is conserved between cofilin and twinfilin. We have now included a new panel (Fig. S10B) showing a full alignment of human cofilin-1 with the N-terminal ADF domain of human twinfilin-1. From the alignment it is evident that cofilin and twinfilin are more similar overall in the LIMK interacting helix than they are across the entire sequence. However, there are several other highly similar regions including those involved in actin binding. In the original Fig. S10B,C we attempted to show that the residues of cofilin that we observed previously to interact with LIMK crystallographically (Hamill et al, 2016, PMID: 27153537) are also found in twinfilin. In these panels we had highlighted all residues that displayed *any* contact with LIMK in our kinase-substrate structure (PDB: 5HVK) as illustrated schematically in Hamill et al., 2016 and reproduced below. However, the reviewer correctly points out that we need to provide some additional context. Some of these residues are extremely important for the interaction, as was verified by mutagenesis in our original study, including Met115 (Met99 in twinfilin). However, other residues such as Asp122 and Lys127 mentioned by the reviewer only make minor contributions to the interface. To clear up this ambiguity and focus the reader on the part of the cofilin that is important for LIMK recognition and well-conserved in twinfilin we have updated Figure S10B,C (now Fig. S10C,D) as a zoomed-in view to show only those critical residues.

Figure panel from Hamill, et al. (2006) showing contacts evident in the LIMK1-cofilin crystal structure (PDB code 5HVK), with interactions shown with dotted lines. Residues mutated in the study, all of which decreased the LIMK-cofilin interaction, are in red.

6. Line 196: *I don't think these spot assays (in Fig 3A-B) can be translated into specific growth rates (e.g., "5-fold"). The spot densities likely reflects a mix of growth rate (affecting colony size) and survival (affecting colony number). It's probably better to just say growth was "slower", rather than claim a specific rate.*

We agree and have modified the text accordingly.

Minor points:

7. Line 191: *"...the "consensus" double mutant cofilin A2L,V5M...". The meaning of "consensus" here is obscure; please explain in what sense this is a consensus.*

We described this as a consensus mutant because it has the most enriched residues at positions 2, 4 and 5. To avoid confusion, we now provide the specific definition and no longer refer to it as a "consensus" mutant.

8. Lines 237 and 240: Fractions don't need superscripts; e.g., 1/3 means one-third, there's no need to add "rd".

Corrected.

9. Line 245: I think this should cite Fig 4B,C (not 4A,B).

Corrected.

10. Lines 257-259: I think this should cite Fig 5C (not 5D). Also, clarify that this result refers to IN VITRO phosphorylation; otherwise, it sounds redundant with the in vivo findings in the preceding sentence describing Fig 5B.

Corrected.

11. Line 269: In current Figure the relevant panel here is marked as 5D, not 5F. Moreover, in the Fig 5 legend the descriptions of panels D/E appear to be labeled as panels F/G.

Corrected.

12. In Fig 7C, the symbol for S3D G4F double mutant is different in the curve (diamond) versus the key (inverted triangle).

Corrected.

Reviewer #3

In this manuscript, the authors have investigated N-terminal mechanisms of cofilin function and phospho-regulation by its cognate kinase, LIMK. They employ budding yeast and an elegant system to screen a randomized library of N-terminal cofilin mutants to identify sequence variants supporting actin regulation/depolymerization (indicated by growth in culture) and, concurrently, phospho-regulation by LIMK1.

The authors make some interesting findings and conclusions with overall outcomes that are broadly consistent with the current understanding of cofilin biology. Enriched variants align with native sequences. Variants that support growth in culture reflect capacities for actin binding and filament severing (with the later intermittently evaluated in this study). Val5 position confers the greatest selectivity for cofilin function and while Gly4 is invariant, this has intermediate selectivity for function. While residue 2 is invariably Ala, this seems to be most dispensable for function. LIMK phosphorylates serine3, can phosphorylate Tyr in this position, but not Thr which has been previously demonstrated as referenced by authors.

Most interestingly, Phe substitution of Gly4 does not prevent Ser3 phosphorylation but removes or negates its regulatory effects on cofilin activity. This is interesting as Ser3 phosphorylation status is widely associated with cofilin1 inactivation/regulation and this observation provides one biochemical context where these events are effectively uncoupled.

The experiments are well executed with convincing datasets that support the various conclusions. However, my main reservation for publication in journal such as Nature Communications is that main findings, as outline above, are either unsurprising or their broad significance in protein phosphoregulation unclear.

We thank the reviewer for their positive comments. We respectfully disagree with the contention that the main findings of the paper are entirely expected. Specific points are addressed below, but we believe our results do provide new insight into the LIMK/cofilin system. For example, we did not anticipate that cofilin N-terminal sequence requirements for function and regulation would be so loose when considered separately,

but that collectively they would conform largely to the native conserved sequence. Furthermore, the decoupling of phosphorylation from inhibition found with some functional mutants could not have been expected and is not reflected in any current models of cofilin phosphoregulation. While these aspects are unique to the LIMK/cofilin system, we believe they do speak to general concepts of how tradeoffs between kinase recognition, regulation, and function are managed in a phosphoregulatory system.

Hydrophobic residues at the position 4 removes phospho-regulation by LIMK is an interesting observation. But what is the broader significance given Gly4 is invariant in nature? The mechanistic basis that uncouples Ser3 phosphorylation from activity regulation is underdetermined but speculated to involve dynamic flexibility of N-terminal region. Whether this represents a generalized mechanism for other ADF/cofilins, LIMK substrates or in protein phosphoregulation more broadly is unclear. I'm also not entirely convinced that this explains the absolute conservation of Gly at 4. The residue at the 4 position has intermediate selectivity for supporting growth and actin regulation so some variability at this position could be expected. The speculated mechanisms in the discussion do not exclude another small residue (Fig 4B) that could similarly be conducive for cofilin function, LIMK1 phosphoregulation and flexibility of N-terminus. Conversely, Asp and Tyr at position 4 are charged or bulky residues that could similarly influence flexibility of the N-terminal region or the charged phosphate on Ser 3 but shown to support growth and are depleted with induction of LIMK (Fig. 4). For completeness, the experiment in Figure 7 should incorporate G4L cofilin variant that is similar phosphorylated and rescues yeast growth in presence of LIMK.

The reviewer raises valid points regarding our ability to explain why some residues are tolerated for cofilin function, in particular at position 4. As indicated, this residue is relatively tolerant to substitution. However, Gly4 substitutions that do support growth but at a slower rate than WT cofilin have reduced fitness and would thus confer an evolutionary disadvantage. From this standpoint, we agree that it is important for us to examine the G4L variant, since Gly and Leu were enriched to the same extent at position 4 in our screen (see heat map in Fig 2B and sequence logo in Fig S3A). Accordingly, we have now conducted yeast growth and in vitro actin binding assays comparing cofilin-G4L with cofilin-S3D/G4L (included in Fig 7A,B and Table 1). These experiments confirm that cofilin-G4L supports yeast growth to the same extent as WT cofilin, in contrast to cofilin-G4F that conferred a slight growth defect. Furthermore, our new experiment clearly shows that like G4F, G4L reverses the inhibitory effect of the phosphomimetic S3D mutation, both in supporting yeast growth and in binding actin. While we cannot rule out other potential reasons for the absolute conservation of Gly4, these data support the idea that at least Leu4 is deselected due to its inability to support phosphoregulation. As the reviewer points out, at present we can only speculate as to the mechanism for this uncoupling. The increased flexibility imparted by Gly4 may contribute, but as we indicate in the discussion, hydrophobic residues could balance repulsion caused by Ser3 phosphorylation through favorable interactions with the hydrophobic pocket of actin.

As phosphomimics do not fully recapitulate phosphorylation chemically or functionally (e.g. Fig 3C), this reviewer is also left wondering whether actin binding/severing is similarly maintained by G4L and G4F mutants that are in vitro phosphorylated by LIMK1 (compared to a phosphomimic). One would expect it to be and easily confirmed given the established proteins and methods?

We agree that ideally we would perform these experiments with authentically phosphorylated cofilin Gly4 mutants due to caveats with the use of phosphomimetics. We had in fact attempted to do this very experiment with cofilin^{G4F} prior to the original submission. However, unlike WT cofilin, we could never separate phospho-cofilin^{G4F} from residual non-phosphorylated cofilin^{G4F} by ion exchange and/or size exclusion chromatography despite substantial efforts (see Figure below for reviewer showing the best of multiple attempts we made to purify phospho-cofilin^{G4F}). Using this partially purified phospho-cofilin^{G4F} we conducted one actin binding experiment, and we did observe binding (see below). We believe that the most likely explanation for this result is that phospho-cofilin^{G4F} maintains some capacity to bind actin. However, given that the preparation included a substantial amount (~20 – 25% of total) of non-phosphorylated cofilin, we could not rule out that the observed actin binding was attributable to the unphosphorylated component. For this reason, we felt it was inappropriate to include the data in our manuscript. We were thus unfortunately limited to including experiments with the phosphomimetic S3D mutation. We concede that it is formally possible that true phosphorylation has a larger inhibitory effect on actin binding that cannot be overcome by Leu4 or Phe4 substitution. However, taken together with our observations that G4F and G4L support growth even when phosphorylated by LIMK in cells, we believe the most straightforward explanation is that those mutants can still bind and sever actin even when phosphorylated.

Reviewer figure: Purification and analysis of phospho-cofilin^{G4F}. Phos-tag gel at left shows substantial residual non-phosphorylated cofilin^{G4F} following purification through MonoS ion exchange and S200 size exclusion chromatography. By contrast, WT cofilin can be purified in almost entirely phosphorylated form. In the pyrene-actin fluorescence quenching assay shown at right, the same phospho-cofilin^{G4F} preparation had an $K_{0.5}$ for approximately 1.5 µM. For comparison, non-phosphorylated cofilin^{G4F} had an $K_{0.5}$ for actin binding of 1.0 (Table 1 in manuscript).

My other suggestion relates to the twinfilin expts which adds incrementally to phosphoacceptor site preferences for LIMK in Fig. 5 but seems also to present an

opportunity to investigate phosphoserine regulation of actin binding/severing properties of this conserved ADF protein (doi.org/10.1242/jcs.02860) and whether the flanking conserved Gly residue at +1 position has similar influence over the phosphoacceptor site as demonstrate in cofilin1.

We agree that regulation of twinfilin by Thr5 phosphorylation would be an interesting avenue to pursue. Indeed, the evolutionary conservation of a phosphorylatable residue at that position, as well as a Gly residue immediately downstream, suggest that it is likely regulated in this way. However, we feel that investigating this phenomenon is a diversion from the main focus of this manuscript on the LIMK-cofilin system. A convincing investigation of twinfilin regulation would require a substantial amount of work, including identifying the relevant kinases, verifying phosphorylation in cells, and examining the impact of phosphorylation on multiple distinct biochemical functions ascribed to twinfilin. While such investigations would indeed make an important contribution to the field, they could not be conducted in a reasonable time frame for revision. While we agree that these are avenues worth pursuing in the future, we feel strongly that they are outside the scope of the current manuscript.

Lastly, the manuscript is well written overall with exception Figure 5C-E are incorrectly labelled in the main text. Page 12 description of LIMK1 phosphorylation Ser3 substituted cofilin references Fig. 5D incorrectly. Similarly, references to twinfillin sequence and in vitro phosphorylation are mislabelled as Fig. 5F and 5G respectively. Figure 5 legends on page 35 also incorrectly indicate figure panels 5D and 5E as F and G.

We thank the reviewer for pointing out these errors, which have now been corrected.

Reviewers' Comments:

Reviewer #2:

Remarks to the Author:

In this revised manuscript, the authors have addressed the (relatively minor) issues raised during the initial review in a thoughtful and satisfactory manner. I have no further concerns; the work is sound and compelling.

Reviewer #3:

Remarks to the Author:

In their revised submission, the authors have made a measured and meaningful attempt to address comments in my initial review including additional experimental work which I appreciate.

They have included the S3D, G4L variant in Figure 7 which also decouples phosphomimetic inhibition. They further acknowledge that in vitro phosphorylation of these variants by LIMK represents an appropriate idealized experiment but this was apparently not straightforward. I accept the reason and appreciate the supporting but preliminary data provided in the reviewer figure.

Lastly, my other comment was whether Fig 7 findings extended to other ADF family proteins. Given identified similarities in twinfillin phospho-recognition by LIMK, this seemed an obvious next step. I accept that I may not have, in my initial review, fully appreciated the scope of the work required to experimentally demonstrate twinfillin phosphorylation and the evasion of negative regulatory effects. I maintain, however, that this would have further elevated the broad significance of paper findings.

That all said, i am supportive of publication of this revised manuscript.